# rStar-Math: Small LLMs Can Master Math Reasoning with Self-Evolved Deep Thinking

Xinyu Guan [* 1 2]  Li Lyna Zhang [* 1 †]  Yifei Liu [3 1]  Ning Shang [1]  Youran Sun [1 4]  Yi Zhu [1]

Fan Yang [1]  Mao Yang [1]

## Abstract

We present rStar-Math to demonstrate that small language models (SLMs) can rival or even surpass the math reasoning capability of OpenAI o1, without distillation from superior models. rStar-Math achieves this by exercising "deep thinking" through Monte Carlo Tree Search (MCTS), where a math *policy SLM* performs test-time search guided by an SLM-based *process reward model*. rStar-Math introduces three innovations to tackle the challenges in training the two SLMs: **(1)** a novel code-augmented CoT data synthesis method, which performs extensive MCTS rollouts to generate *step-by-step verified reasoning trajectories* used to train the policy SLM; **(2)** a novel process reward model training method that avoids naïve step-level score annotation, yielding a more effective *process preference model (PPM)*; **(3)** a *self-evolution recipe* in which the policy SLM and PPM are built from scratch and iteratively evolved to improve reasoning capabilities. Through 4 rounds of self-evolution with millions of synthesized solutions for 747k math problems, rStar-Math boosts SLMs' math reasoning to state-of-the-art levels. On MATH benchmark, it improves Qwen2.5-Math-7B from 58.8% to 90.0%, surpassing o1-preview by +4.5%. On the USA Math Olympiad (AIME), rStar-Math solves an average of 53.3% (8/15) of problems, ranking among the top 20% of the brightest high school math students. Code and data are available at https://github.com/microsoft/rStar.

---

[*]Equal contribution  [1]Microsoft Research Asia [2]Peking University [3]University of Science and Technology of China [4]Tsinghua University; Xinyu Guan, Yifei Liu and Youran Sun did this work during the internship at MSRA. Correspondence to: Li Lyna Zhang <lzhani@microsoft.com>.

*Proceedings of the 42nd International Conference on Machine Learning*, Vancouver, Canada. PMLR 267, 2025. Copyright 2025 by the author(s).

| Task (pass@1 Acc) | rStar-Math (Qwen-7B) | rStar-Math (Phi3-mini) | OpenAI o1-preview | OpenAI o1-mini | GPT-4o | DeepSeek-V3 |
|---|---|---|---|---|---|---|
| MATH | 90.0 | 88.6 | 85.5 | 90.0 | 76.6 | 90.2 |
| AIME 2024 | 53.3 | 46.7 | 44.6 | 56.7 | 9.3 | 39.2 |
| Olympiad Bench | 65.6 | 60.3 | - | 65.3 | 43.3 | 55.4 |
| College Math | 60.5 | 59.3 | - | 57.8 | 48.5 | 58.9 |
| Omni-Math | 50.5 | 48.5 | 52.5 | 60.5 | 49.6 | 35.9 |

*Table 1.* rStar-Math enables frontier math reasoning in SLMs via deep thinking over 64 trajectories.

## 1 Introduction

Recent studies have demonstrated that large language models (LLMs) are capable of tackling mathematical problems (Team, 2024; Yang et al., 2024; Liu et al., 2024). However, the conventional approach of having LLMs generate complete solutions in a single inference – akin to System 1 thinking (Daniel, 2011) – often yields fast but error-prone results (Valmeekam et al., 2023; OpenAI, 2023). In response, test-time compute scaling (Snell et al., 2024; Qi et al., 2024) suggests a paradigm shift toward a System 2-style thinking, which emulates human reasoning through a slower and deeper thought process. In this paradigm, an LLM serves as a policy model to generate multiple math reasoning steps, which are then evaluated by another LLM acting as a reward model (OpenAI, 2024). The steps and solutions deemed more likely to be correct are selected. The process repeats iteratively and ultimately derives the final answer.

In the test-time compute paradigm, the key is to train a powerful policy model that generates promising solution steps and a reliable reward model that accurately evaluates them, both of which depend on *high-quality* training data. Unfortunately, it is well-known that off-the-shelf high-quality math reasoning data is scarce, and synthesizing high-quality math data faces fundamental challenges. For the policy model, it is challenging to distinguish erroneous reasoning steps from the correct ones, complicating the elimination of low-quality data. It is worth noting that in math reasoning, a correct final answer does not ensure the correctness of the entire reasoning trace (Lanham et al., 2023). Incorrect intermediate steps significantly decrease data quality. As for the reward model, process reward modeling (PRM) shows a great potential by providing fine-grained feedback on intermediate steps (Lightman et al., 2023). However, the training data is even scarcer in this regard: accurate step-by-step feedback

requires intense human labeling efforts and is impractical to scale, while those automatic annotation attempts show limited gains due to noisy reward scores (Luo et al., 2024; Chen et al., 2024). Due to the above challenges, existing distill-based data synthesis approaches to training policy models, e.g., scaling up GPT4-distilled CoT data (Tang et al., 2024; Huang et al., 2024), have shown diminishing returns and cannot exceed the capability of their teacher model; meanwhile, as of today, training reliable PRMs for math reasoning remains an open question.

In this work, we introduce **rStar-Math**, a self-evolvable System 2-style reasoning approach that achieves the state-of-the-art math reasoning, rivaling and sometimes even surpassing OpenAI o1 on challenging math competition benchmarks with a model size as small as 7 billion. Unlike solutions relying on superior LLMs for data synthesis, rStar-Math leverages smaller language models (SLMs) with Monte Carlo Tree Search (MCTS) to establish a self-evolutionary process, iteratively generating higher-quality training data. To achieve self-evolution, rStar-Math introduces three key innovations.

First, a novel code-augmented CoT data synthesis method, which performs *extensive* MCTS rollouts to generate *step-by-step verified reasoning trajectories* with *self-annotated MCTS Q-values*. Specifically, math problem-solving is decomposed into multi-step generation within MCTS. At each step, the SLM serving as the policy model samples candidate nodes, each generating a one-step CoT and the corresponding Python code. To verify the generation quality, only nodes with successful Python code execution are retained, thus mitigating errors in intermediate steps. Moreover, extensive MCTS rollouts automatically assign a Q-value to each intermediate step based on its contribution: steps contributing to more trajectories that lead to the correct answer are given higher Q-values and considered higher quality. This ensures that the reasoning trajectories generated by SLMs consist of correct, high-quality intermediate steps.

Second, a novel method that trains an SLM acting as a *process preference model*, i.e., a PPM to implement the desired PRM, that reliably predicts a reward label for each math reasoning step. The PPM leverages the fact that, although Q-values are still not precise enough to score each reasoning step despite using extensive MCTS rollouts, the Q-values can reliably distinguish positive (correct) steps from negative (irrelevant/incorrect) ones. Thus the training method constructs preference pairs for each step based on Q-values and uses a pairwise ranking loss (Ouyang et al., 2022) to optimize PPM's score prediction for each reasoning step, achieving reliable labeling. This approach avoids conventional methods that directly use Q-values as reward labels (Luo et al., 2024; Chen et al., 2024), which are inherently noisy and imprecise in stepwise reward assignment.

Finally, a four-round self-evolution recipe that progressively builds both a frontier policy model and PPM from scratch. We begin by curating a dataset of 747k math word problems from publicly available sources. In each round, we use the latest policy model and PPM to perform MCTS, generating increasingly high-quality training data using the above two methods to train a stronger policy model and PPM for next round. Each round achieves progressive refinement: (1) a stronger policy SLM, (2) a more reliable PPM, (3) generating better reasoning trajectories via PPM-augmented MCTS, and (4) improving training data coverage to tackle more challenging competition-level math problems.

Extensive experiments across four SLMs (1.5B-7B) and seven math reasoning tasks demonstrate the effectiveness of rStar-Math. Remarkably, rStar-Math improves all four SLMs, matching or even surpassing OpenAI o1 on challenging math benchmarks. On MATH benchmark, with 8 search trajectories, rStar-Math boosts Qwen2.5-Math-7B from 58.8% to 89.4% and Qwen2.5-Math-1.5B from 51.2% to 87.8%. With 64 trajectories, the scores rise to 90% and 88.4%, outperforming o1-preview by 4.5% and 2.6% and matching o1-mini's 90%. On the Olympiad-level AIME 2024, rStar-Math solves on average 53.3% (8/15) of the problems, exceeding o1-preview by 8.7% and all other open-sourced LLMs. We further conduct comprehensive experiments to verify the superiority of step-by-step verified reasoning trajectories over state-of-the-art data synthesis baselines, as well as the PPM's effectiveness compared to outcome reward models and Q value-based PRMs.

## 2 Related Works

**Math Data Synthesis**. Advancements in LLM math reasoning have largely relied on curating high-quality CoT data, with most leading approaches being GPT-distilled, using frontier models like GPT-4 for synthesis (Wang et al., 2024b; Gou et al., 2023; Luo et al., 2023). Notable works include NuminaMath (Jia LI & Polu, 2024a) and MetaMath (Yu et al., 2023b). While effective, this limits reasoning to the capabilities of the teacher LLM. Hard problems that the teacher LLM cannot solve are excluded in the training set. Even solvable problems may contain error-prone intermediate steps, which are hard to detect. Although rejection sampling methods (Yuan et al., 2023; Brown et al., 2024) can improve data quality, they do not guarantee correct intermediate steps. As a result, naively scaling up CoT data generation using frontier LLMs yields diminishing returns, with performance gains approaching saturation.

**Scaling Test-time Compute** has introduced new scaling laws, allowing LLMs to improve performance across by generating multiple samples and using reward models for best-solution selection (Snell et al., 2024; Wu et al., 2024; Brown et al., 2024). Various search methods have been proposed (Kang et al., 2024; Wang et al., 2024a), including

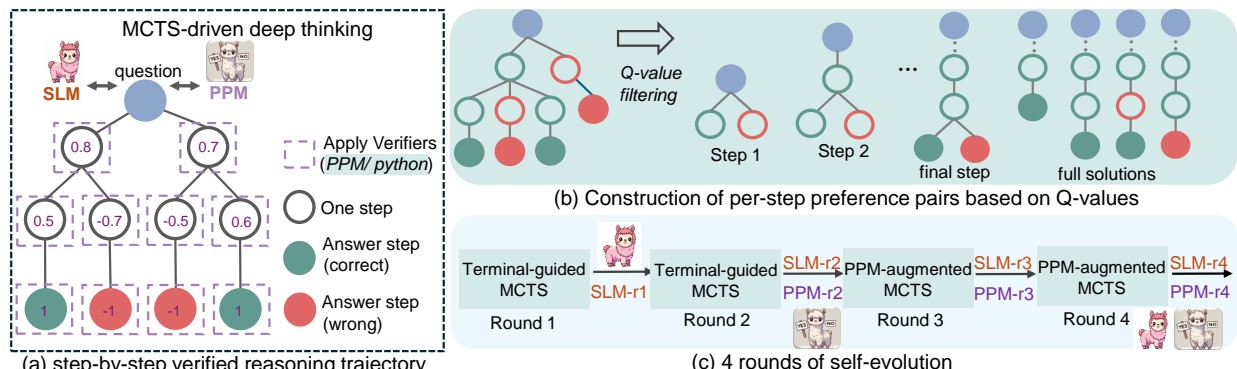

*Figure 1.* The overview of rStar-Math.

random sampling (Wang et al., 2023) and tree-search methods (Yao et al., 2024; Zhang et al., 2024a; Qi et al., 2024) like MCTS. However, open-source methods for scaling test-time compute have shown limited gains in math reasoning, often due to policy LLM or reward model limitations.

**Reward Models** are crucial for effective System 2 reasoning but are challenging to obtain. The major works include Outcome Reward Model (Yang et al., 2024; Yu et al., 2023a) and Process Reward Model (PRM) (Lightman et al., 2024). While PRMs offer promising dense, step-level reward signals for complex reasoning (Luo et al., 2024), collecting step-level annotations remains an obstacle. While (Kang et al., 2024; Wang et al., 2024a) rely on costly human-annotated datasets like PRM800k (Lightman et al., 2024), recent approaches (Wang et al., 2024c; Luo et al., 2024) explore automated annotation via Monte Carlo Sampling or MCTS. However, they struggle to generate precise reward scores, limiting performance gains. rStar-Math introduces a novel process preference reward (PPM) that eliminates the need for accurate step-level reward score annotation.

## 3 Methodology

In our work, we explore using two 7B SLMs to generate higher-quality training data. However, self-generating data presents significant challenges for SLMs. SLMs frequently fail to generate correct solutions, and even when the final answer is correct, the intermediate steps are often flawed or of poor quality. Moreover, SLMs solve fewer challenging problems compared to advanced models like GPT-4.

This section introduces our methodology, as illustrated in Fig. 1. To mitigate errors and low-quality intermediate steps, we introduce a code-augmented CoT synthetic method, which performs extensive MCTS rollouts to generate step-by-step verified reasoning trajectories, annotated with Q-values. To further improve SLM performance on challenging problems, we introduce a four-round self-evolution recipe. In each round, both the policy SLM and the reward model are updated to stronger versions, progressively tackling more difficult problems and generating higher-quality

training data. Finally, we present a novel process reward model training approach that eliminates the need for precise per-step reward annotations, yielding the more effective process preference model (PPM).

### 3.1 Step-by-Step Verified Reasoning Trajectory

Given a problem $x$ and a policy model $M$, we use standard MCTS to incrementally construct a search tree for step-by-step solution exploration. As shown in Fig. 1(a), the root node represents question $x$, and child nodes correspond to intermediate steps $s$ generated by $M$. A root-to-leaf path ending at terminal node $s_d$ forms a trajectory $\mathbf{t} = x \oplus s_1 \oplus s_2 \oplus ... \oplus s_d$, with each step $s_i$ assigned a Q-value $Q(s_i)$. From the search tree $\mathcal{T}$, we extract solution trajectories $\mathbb{T} = \{\mathbf{t}^1, \mathbf{t}^2, ..., \mathbf{t}^n\}(n \geq 1)$. Our goal is to select high-quality trajectories from $\mathcal{T}$ to construct the training set. To achieve this, we introduce code-augmented CoT synthesis method to filter out low-quality generations and perform extensive rollouts to improve the reliability of Q-value accuracy.

**Code-augmented CoT Generation**. Prior MCTS approaches primarily generate natural language (NL) CoTs (Qi et al., 2024; Zhang et al., 2024b). However, LLMs often suffer from hallucination, producing incorrect or irrelevant steps yet still arrive at the correct answer by chance (Lanham et al., 2023). These flawed steps are challenging to detect and eliminate. To address this, we propose a novel code execution augmented CoT. As shown in Fig. 2, the policy model generates a one-step NL CoT alongside its corresponding Python code, where the NL CoT is embedded as a Python comment. Only generations with successfully executed Python code are retained as valid candidates.

Specifically, starting from the root node $x$, we perform multiple MCTS iterations through *selection*, *expansion*, *simulation*, and *back-propagation*. At step $i$, we collect the latest reasoning trajectory $x \oplus s_1 \oplus s_2 \oplus ... \oplus s_{i-1}$ as the current state. Based on this state, we prompt (see Appendix A.4) the policy model to generate $n$ candidates $s_{i,0}, ..., s_{i,n-1}$. Python code execution is then employed to filter valid nodes. As shown in Fig. 2, each generation

**Question:** Bill walks $\frac{1}{2}$ mile south, then $\frac{3}{4}$ mile east, and finally $\frac{1}{2}$ mile south. How many miles is he, in a direct line, from his starting point? Express your answer as a decimal to the nearest hundredth.

```
# Step 1: Calculate the total distance walked south
total_south = 1/2 +1/2
# Step 2: Calculate the total distance walked east
total_east = 3/4
# Step 3: Use the Pythagorean theorem to find the direct distance from the starting point
import math
direct_distance = math.sqrt(total_south**2 + total_east**2)
# Step 4: Round the direct distance to the nearest hundredth
direct_distance_rounded = round(direct_distance, 2
From the result, we can see that the direct distance from the starting point is \boxed{1.25} miles
```

➡ NL CoT as Python Comment

*Python code execution for step 1:*
```
# Step 1: Calculate the total distance walked south
total_south = 1/2 +1/2
```

*Python code execution for step 2:*
```
# Step 1: Calculate the total distance walked south
total_south = 1/2 +1/2
# Step 2: Calculate the total distance walked east
total_east = 3/4
...
```

*Figure 2.* An example of Code-augmented CoT. To avoid syntax errors, the code at step $i$ is executed by concatenating it with code from all previous steps. For example, step 2 includes code from step 1 to ensure correct execution.

$s_{i,j}$ is concatenated with the code from all previous steps, forming $s_1 \oplus s_2 \oplus ... \oplus s_{i-1} \oplus s_{i,j}$. Candidates that execute successfully are retained as valid nodes and scored by the PPM, which assigns a Q-value $q(s_i)$. Then, we use the well-known Upper Confidence bounds for Trees (UCT) (Kocsis & Szepesvári, 2006) to select the best node among the $n$ candidates. This selection is mathematically represented as:

$$\text{UCT}(s) = Q(s) + c\sqrt{\frac{\ln N_{parent}(s)}{N(s)}}; \quad Q(s) = \frac{q(s)}{N(s)} \quad (1)$$

where $N(s)$ denotes the number of visits to node $s$, and $N_{\text{parent}}(s)$ is the visit count of $s$'s parent node. The predicted reward $q(s)$ is provided by the PPM and will be updated through back-propagation. $c$ is a constant that balances exploitation and exploration.

We repeat the process until reaching a terminal node, either by arriving a final answer or hitting the maximum tree depth. This is referred to as a *rollout*. Based on whether the rollout reaches the correct answer, we perform back-propagation to update the trajectory Q-value scores. We introduce the score annotation methods in next sections.

**Extensive Rollouts for Q-value Annotation**. Accurate Q-value $Q(s)$ annotation in Eq. 1 is crucial for guiding MCTS towards correct problem-solving paths and identifying high-quality steps within trajectories. Following AlphaGo (Silver et al., 2017) and rStar (Qi et al., 2024), we perform MCTS rollout to assign a Q-value to each step. However, insufficient rollouts can lead to spurious Q-value assignments, such as overestimating suboptimal steps. To mitigate this issue, we draw inspiration from Go players, who refine their evaluations for each move through repeated gameplay. Analogously, within each rollout, we update the Q-value of each step based on its contribution to achieving the correct final answer. Through extensive MCTS rollouts, steps consistently leading to correct answers achieve higher Q-values, occasional successes yield moderate Q-values, and consistently incorrect steps receive low Q-values. Specifically, we introduce two self-annotation methods to obtain these step-level Q-values. Fig. 1(c) shows the detailed setting in the four rounds of self-evolution.

*Terminal-guided annotation*. During the first two rounds, when the PPM is unavailable or insufficiently accurate, we use terminal-guided annotation. Formally, let $q(s_i)^k$ denote the q value for step $s_i$ after back-propagation in the $k^{th}$ rollout, we score per-step Q-value as follows:

$$q(s_i)^k = q(s_i)^{k-1} + q(s_d)^k; \quad (2)$$

where the initial q value $q(s_i)^0 = 0$ in the first rollout. If this step frequently leads to a correct answer, its $q$ value will increase; otherwise, it decreases. Terminal nodes (denoted as $s_d$ in Eq. 2) are scored as $q(s_d) = 1$ for correct answers and $q(s_d) = -1$ otherwise, as shown in Fig. 1.

*PPM-augmented annotation*. Starting from the third round, we use PPM to score each step for more effective generation. Compared to terminal-guided annotation, which requires multiple rollouts for a meaningful $q$ value, PPM directly predicts a non-zero initial $q$ value. PPM also helps the policy model to generate higher-quality steps, guiding solutions towards correct paths. Formally, for step $s_i$, PPM predicts an initial $q(s_i)^0$ value based on the partial trajectory:

$$q(s_i)^0 = PPM(x \oplus s_1 \oplus s_2 \oplus ... \oplus s_{i-1} \oplus s_i) \quad (3)$$

This $q$ value will be updated based on terminal node's $q(s_d)$ value through MCTS *back-propagation* in Eq. 2. For terminal node $s_d$, we do not use PPM for scoring during training data generation. Instead, we assign a more accurate score based on ground truth labels as terminal-guided rewarding.

### 3.2  Process Preference Model

Process reward models, which provide granular step-level reward signals, is highly desirable for solving challenging math problems. However, obtaining high-quality step-level training data remains an open challenge. Existing methods rely on human annotations (Lightman et al., 2023) or MCTS-generated scores (Zhang et al., 2024b; Chen et al., 2024) to label a score for each step. These scores then serve as training targets, with methods such as MSE loss (Chen et al., 2024) or pointwise loss (Wang et al., 2024c; Luo et al., 2024; Zhang et al., 2024b) used to minimize the difference between predicted and labeled scores. As a result, the precision of these annotated step-level scores determines the effectiveness of the final process reward model.

Unfortunately, precise per-step scoring remains a unsolved challenge. Although our extensive MCTS rollouts improve the reliability of Q-values, precisely evaluating fine-grained step quality presents a major obstacle. For instance, among a set of correct steps, it is difficult to rank them as best, second-best, or average and then assign precise scores. Similarly, among incorrect steps, differentiating the worst from moderately poor steps poses analogous challenges. Even expert human annotation struggles with consistency, particularly at scale, leading to inherent noise in training labels.

We introduce a novel training method that trains a process preference model (PPM) by constructing step-level positive-negative preference pairs. As shown in Fig. 1(b), instead of using Q-values as direct reward labels, we use them to select steps from MCTS tree for preference pair construction. For each step, we select two candidates with the highest Q-values as positive steps and two with the lowest as negative steps. Critically, the selected positive steps must lead to a correct final answer, while negative steps must lead to incorrect answers. For intermediate steps (except the final answer step), the positive and negative pairs share the same preceding steps. For the final answer step, where identical reasoning trajectories rarely yield different final answers, we relax this restriction. We select two correct trajectories with the highest average Q-values as positive examples and two incorrect trajectories with the lowest average Q-values as negative examples. Following (Ouyang et al., 2022), we define our loss function using the standard Bradley-Terry model with a pairwise ranking loss:

$$\mathcal{L}_{ppm}(\theta) = -\frac{1}{4} E_{(x,y_i^{pos},y_i^{neg} \in \mathbb{D})}[log(\sigma(r_\theta(x, y_i^{pos}) - r_\theta(x, y_i^{neg})))]$$

when $i$ is not final answer step, $\begin{cases} y_i^{pos} = s_1 \oplus ... \oplus s_{i-1} \oplus s_i^{pos}; \\ y_i^{neg} = s_1 \oplus ... \oplus s_{i-1} \oplus s_i^{neg} \end{cases}$ (4)

$r_\theta(x, y_i)$ denotes the PPM output, where $x$ is the problem and $y$ is the trajectory from the first step to the $i^{th}$ step. $\sigma$ denotes the sigmoid function.

### 3.3 Self-Evolved Deep Thinking

**Math Problems Collection**. We collect a large dataset of 747k math word problems with ground-truth answers, primarily from NuminaMath (Jia LI & Polu, 2024a) and MetaMath (Yu et al., 2023b). Notably, only competition-level problems (e.g., Olympiads and AIME/AMC) from NuminaMath are included, as we observe that grade-school-level problems do not significantly improve LLM complex math reasoning. To augment the limited competition-level problems, we follow (Li et al., 2024) and use GPT-4 to synthesize new problems based on the seed problems in 7.5k MATH train set and 3.6k AMC-AIME training split. However, GPT-4 often generated unsolvable problems or incorrect solutions for challenging seed problems. To filter these, we prompt GPT-4 to generate 10 solutions per problem, retaining only those with at least 3 consistent solutions.

*Table 2.* Percentage of the 747k math problems correctly solved in each round. The first round uses DeepSeek-Coder-Instruct as the policy LLM, while later rounds use our fine-tuned 7B policy SLM.

| # | models in MCTS | GSM-level | MATH-level | Olympiad-level | All |
|---|---|---|---|---|---|
| Round 1 | DeepSeek-Coder-V2-Instruct | 96.61% | 67.36% | 20.99% | 60.17% |
| Round 2 | policy SLM-r1 | 97.88% | 67.40% | 56.04% | 66.60% |
| Round 3 | policy SLM-r2, PPM-r2 | 98.15% | 88.69% | 62.16% | 77.86% |
| Round 4 | policy SLM-r3, PPM-r3 | 98.15% | 94.53% | 80.58% | 90.25% |

**Reasoning Trajectories Collection**. Instead of using the original solutions in the 747k math dataset, we conduct extensive MCTS rollouts (Sec. 3.1) to generate higher-quality step-by-step verified reasoning trajectories. In each self-evolution round, we perform 16 rollouts per math problem, which leads to 16 reasoning trajectories. Problems are then categorized by difficulty based on the correct ratio of the generated trajectories: *easy* (all solutions are correct), *medium* (a mix of correct and incorrect solutions) and *hard* (all solutions are incorrect). For *hard* problems with no correct trajectories, an additional MCTS with 16 rollouts is performed. After that, all step-by-step trajectories and their annotated Q-values are collected and filtered to train the policy SLM and process preference model.

**Self-evolution Recipe**. Due to the weaker capabilities of SLMs, we perform four rounds of self-evolution to progressively generate higher-quality data. Each round uses MCTS to generate step-by-step verified reasoning trajectories, which are then used to train the new policy SLM and PPM. The new models are then used in next round. Fig. 1(c) and Table 2 detail the models used for data generation in each round, along with the new trained models. Next, we outline the specific improvements targeted in each round.

◇ *Round 1: Bootstrapping an initial strong policy SLM-r1*. To enable SLMs to self-generate reasonably good training data, we perform a bootstrap round to fine-tune an initial strong policy model. As shown in Table 2, we run MCTS with DeepSeek-Coder-V2-Instruct (236B) to collect the SFT data. With no available reward model in this round, we use terminal-guided annotation for Q-values and limit MCTS to 8 rollouts for efficiency. For correct solutions, the top-2 trajectories with the highest average Q-values are selected as SFT data, which are then used to SFT the first policy SLM, denoted as SLM-r1.

◇ *Round 2: Training a reliable PPM-r2*. In this round, with the policy model updated to the 7B SLM-r1, we conduct extensive MCTS rollouts for more reliable Q-value annotation and train the first reliable reward model, PPM-r2. Specifically, we perform 16 MCTS rollouts per problem. The resulting step-by-step verified reasoning trajectories show significant improvements in both quality and Q-value precision. As shown in Appendix Table 10, PPM-r2 is notably more effective than in the bootstrap round.

◇ *Round 3: PPM-augmented MCTS to significantly improve data quality*. With the reliable PPM-r2, we perform

PPM-augmented MCTS in this round to generate data, leading to significantly higher-quality trajectories. The generated reasoning trajectories and annotated Q-values are then used to train the new policy SLM-r3 and PPM-r3, both of which show significant improvements.

◇ **Round 4: Solving challenging problems**. After the third round, while grade school and MATH problems achieve high success rates, only 62.16% of Olympiad-level problems are included in the training set. This is *NOT* solely due to weak capabilities in our SLMs, as many Olympiad problems remain unsolved by GPT-4 or o1. To improve coverage, we adopt a straightforward strategy. For unsolved problems after 16 rollouts, we perform an additional 64 rollouts, and if needed, increase to 128. We also conduct multiple MCTS tree expansions with different random seeds. This boosts the success rate of Olympiad-level problems to 80.58%.

After four rounds of self-evolution, 90.25% of the 747k math problems are covered in the training set , as shown in Table 2. Among the remaining unsolved problems, most are synthetic. A manual review of a random sample of 20 problems reveals that 19 are incorrectly labeled. Thus, we conclude that the unsolved problems are of low quality and stop the self-evolution at round 4.

## 4 Evaluation

### 4.1 Setup

**Evaluation Datasets**. We evaluate rStar-Math on diverse mathematical benchmarks. In addition to the widely-used GSM8K (Cobbe et al., 2021), we include challenging benchmarks from multiple domains: *(i)* competition and Olympiad-level benchmarks, such as MATH-500 (Lightman et al., 2023), AIME 2024 (AI-MO, 2024a), AMC 2023 (AI-MO, 2024b) and Olympiad Bench (He et al., 2024). Specifically, AIME is the exams designed to challenge the brightest high school math students in America; *(ii)* college-level math problems from College Math (Tang et al., 2024) and *(iii)* out-of-domain math benchmark: GaoKao (Chinese College Entrance Exam) En 2023 (Liao et al., 2024).

**Base Models and Setup**. We use SLMs of different sizes as the base policy models: Qwen2.5-Math-1.5B (Qwen, 2024b), Phi3-mini-Instruct (3B) (Abdin et al., 2024), Qwen2-Math-7B (Qwen, 2024a) and Qwen2.5-Math-7B (Qwen, 2024c).Due to limited GPU resources, we performed 4 rounds of self-evolution exclusively on Qwen2.5-Math-7B, yielding 4 evolved policy SLMs (Table 9) and 4 PPMs (Table 10). For the other 3 policy models, we fine-tune them using step-by-step verified trajectories generated from Qwen2.5-Math-7B's 4th round. PPM from this round is then used as the reward model for the 3 policy SLMs.

**Baselines**. We compare against three strong baselines representing both System 1 and System 2 approaches: **(i)** *Fron-*

*tier LLMs*, including GPT-4o, the latest Claude, OpenAI o1-preview and o1-mini. We measure their accuracy on AMC 2023, Olympiad Bench, College Math, Gaokao and GSM8K, with accuracy numbers for other benchmarks are taken from public technical reports (Team, 2024). **(ii)** *Open-sourced superior reasoning models*; **(iii)** *Both System 1 and System 2 performance of the base models trained from the original models teams*, including Instruct versions (e.g., Qwen2.5-Math-7B-Instruct) and Best-of-N (e.g., Qwen2.5-Math-72B-Instruct+Qwen2.5-Math-RM-72B). Notably, the reward model used for the three Qwen base models is a 72B ORM, significantly larger than our 7B PPM.

**Evaluation Metric**. We report Pass@1 accuracy for all baselines. For System 2 baselines, we use default evaluation settings, such as default thinking time for o1-mini and o1-preview. For Qwen models with Best-of-N, we re-evaluate MATH-500 and AIME/AMC accuracy; other benchmarks results are from their technical reports. For a fair comparison, rStar-Math run MCTS to generate the same number of solutions as Qwen. Specifically, we generate 16 trajectories for AIME/AMC and 8 for other benchmarks, using PPM to select the best solution. We also report performance with increased test-time computation using 64 trajectories, denoted as rStar-Math[64].

### 4.2 Main Results

**Results on diverse challenging math benchmarks**. Table 3 shows the results of rStar-Math with comparing to state-of-the-art reasoning models. We highlight two key observations: **(1)** rStar-Math significantly improves SLMs math reasoning capabilities, achieving performance comparable to or surpassing OpenAI o1 with substantially smaller model size (1.5B-7B). For example, Qwen2.5-Math-7B, originally at 58.8% accuracy on MATH, improved dramatically to 90.0% with rStar-Math, outperforming o1-preview and Claude 3.5 Sonnet while matching o1-mini. On the College Math benchmark, rStar-Math exceeds o1-mini by 2.7%. On AIME 2024, rStar-Math scored 53.3%, ranking just below o1-mini, with the 7B model solving 8/15 problems in both AIME I and II, placing in the top 20% of the brightest high school math students. Notably, 8 of the unsolved problems were geometry-based, requiring visual understanding, a capability rStar-Math currently does not support. **(2)** Despite using smaller policy models (1.5B-7B) and reward models (7B), rStar-Math significantly outperforms state-of-the-art System 2 baselines. Compared to Qwen Best-of-N baselines, which use the same base models (Qwen2-Math-7B, Qwen2.5-Math-1.5B/7B) but a 10× larger reward model (Qwen2.5-Math-RM-72B), rStar-Math consistently improves the reasoning accuracy of all base models to state-of-the-art levels. Even against Best-of-N with a 10× larger Qwen2.5-Math-72B-Instruct policy model, rStar-Math surpasses it on all benchmarks except

*Table 3.* The results of rStar-Math and other frontier LLMs on the most challenging math benchmarks. rStar-Math reports Pass@1 accuracy when searching 16 trajectories for AIME/AMC and 8 for others. rStar-Math[64] shows results when sampling 64 trajectories.

| Model | Method | Competition and College Level | | | | | | OOD |
| | | MATH | AIME 2024 | AMC 2023 | Olympiad Bench | College Math | GSM8K | GaokaoEn 2023 |
|---|---|---|---|---|---|---|---|---|
| *Frontier LLMs* | | | | | | | | |
| GPT-4o | System 1 | 76.6 | 9.3 | 47.5 | 43.3 | 48.5 | 92.9 | 67.5 |
| Claude3.5-Sonnet | System 1 | 78.3 | 16.0 | - | - | - | 96.4 | - |
| GPT-o1-preview | - | 85.5 | 44.6 | 90.0 | - | - | - | - |
| GPT-o1-mini | - | **90.0** | **56.7** | **95.0** | **65.3** | 57.8 | 94.8 | 78.4 |
| *Open-Sourced Reasoning LLMs* | | | | | | | | |
| DeepSeek-Coder-V2-Instruct | System 1 | 75.3 | 13.3 | 57.5 | 37.6 | 46.2 | 94.9 | 64.7 |
| NuminaMath-72B-CoT | System 1 | 64.0 | 3.3 | 70.0 | 32.6 | 39.7 | 90.8 | 58.4 |
| LLaMA3.1-70B-Instruct | System 1 | 65.4 | 23.3 | 50.0 | 27.7 | 42.5 | 94.1 | 54.0 |
| Qwen2.5-Math-72B-Instruct | System 1 | 85.6 | 30.0 | 70.0 | 49.0 | 49.5 | 95.9 | 71.9 |
| Qwen2.5-Math-72B-Instruct+72B ORM | System 2 | 85.8 | 36.7 | 72.5 | 54.5 | 50.6 | 96.4 | 76.9 |
| *General Base Model: Phi3-mini-Instruct (3.8B)* | | | | | | | | |
| Phi3-mini-Instruct (base model) | System 1 | 41.4 | 3.33 | 7.5 | 12.3 | 33.1 | 85.7 | 37.1 |
| **rStar-Math (3.8B SLM+7B PPM)** | System 2 | **85.4** | **40.0** | **77.5** | **59.3** | **58.0** | 94.5 | **77.1** |
| **rStar-Math[64] (3.8B SLM+7B PPM)** | System 2 | **86.4** | **43.3** | **80.0** | **60.3** | **59.1** | 94.7 | **77.7** |
| *Math-Specialized Base Model: Qwen2.5-Math-1.5B* | | | | | | | | |
| Qwen2.5-Math-1.5B (base model) | System 1 | 51.2 | 0.0 | 22.5 | 16.7 | 38.4 | 74.6 | 46.5 |
| Qwen2.5-Math-1.5B-Instruct | System 1 | 60.0 | 10.0 | 60.0 | 38.1 | 47.7 | 84.8 | 65.5 |
| Qwen2.5-Math-1.5B-Instruct+72B ORM | System 2 | 83.4 | 20.0 | 72.5 | 47.3 | 50.2 | 94.1 | 73.0 |
| **rStar-Math (1.5B SLM+7B PPM)** | System 2 | **87.8** | **46.7** | **80.0** | **63.5** | **59.0** | 94.3 | **77.7** |
| **rStar-Math[64] (1.5B SLM+7B PPM)** | System 2 | **88.6** | **46.7** | **85.0** | **64.6** | **59.3** | 94.8 | **79.5** |
| *Math-Specialized Base Model: Qwen2-Math-7B* | | | | | | | | |
| Qwen2-Math-7B (base model) | System 1 | 53.4 | 3.3 | 25.0 | 17.3 | 39.4 | 80.4 | 47.3 |
| Qwen2-Math-7B-Instruct | System 1 | 73.2 | 13.3 | 62.5 | 38.2 | 45.9 | 89.9 | 62.1 |
| Qwen2-Math-7B-Instruct+72B ORM | System 2 | 83.4 | 23.3 | 62.5 | 47.6 | 47.9 | **95.1** | 71.9 |
| **rStar-Math (7B SLM+7B PPM)** | System 2 | **88.2** | **43.3** | **80.0** | **63.1** | **58.4** | 94.6 | **78.2** |
| **rStar-Math[64] (7B SLM+7B PPM)** | System 2 | **88.6** | **46.7** | **85.0** | **63.4** | **59.3** | 94.8 | **79.2** |
| *Math-Specialized Base Model: Qwen2.5-Math-7B* | | | | | | | | |
| Qwen2.5-Math-7B (base model) | System 1 | 58.8 | 0.0 | 22.5 | 21.8 | 41.6 | 91.6 | 51.7 |
| Qwen2.5-Math-7B-Instruct | System 1 | 82.6 | 6.0 | 62.5 | 41.6 | 46.8 | 95.2 | 66.8 |
| Qwen2.5-Math-7B-Instruct+72B ORM | System 2 | 88.4 | 26.7 | 75.0 | 49.9 | 49.6 | **97.9** | 75.1 |
| **rStar-Math (7B SLM+7B PPM)** | System 2 | **89.4** | **50.0** | **87.5** | **65.3** | **59.0** | 95.0 | **80.5** |
| **rStar-Math[64] (7B SLM+7B PPM)** | System 2 | **90.0** | **53.3** | **87.5** | **65.6** | **60.5** | 95.2 | **81.3** |

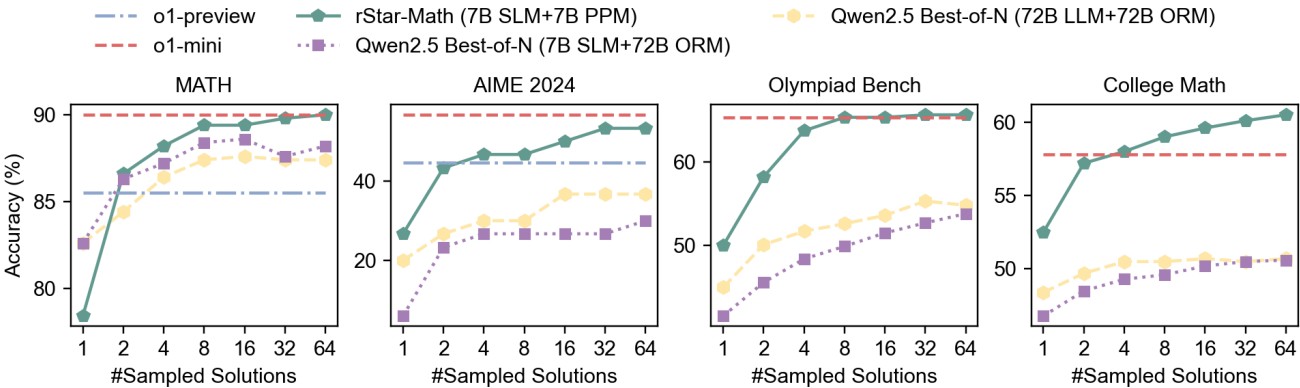

*Figure 3.* Reasoning performance under scaling up the test-time compute.

GSM8K, using the same number of sampled solutions.

**Scaling up test-time computation**. By increasing test-time computation, rStar-Math can explore more trajectories, potentially improving performance. In Fig. 3, we show the impact of test-time compute scaling by comparing the accuracy of the official Qwen Best-of-N across different numbers of sampled trajectories on four challenging math benchmarks. Sampling only one trajectory corresponds to the policy LLM's Pass@1 accuracy, indicating a fallback to System 1 reasoning. We highlight two key observations: **(1)** With only 4 trajectories, rStar-Math significantly outperforms Best-of-N baselines, exceeding o1-preview and approaching o1-mini, demonstrating its effectiveness. **(2)** Scaling test-time compute improves reasoning accuracy across all benchmarks, though with varying trends. On Math, AIME, and Olympiad Bench, rStar-Math shows saturation or slow improvement at 64 trajectories, while on College Math, performance continues to improve steadily.

### 4.3 Ablation Study and Analysis

*Table 4.* The continuously improved math reasoning capabilities via rStar-Math self-evolved deep thinking. Starting from round 2, the 7B base model powered by rStar-Math surpasses GPT-4o.

| Round# | MATH | AIME | AMC | Olympiad Bench | College Math | GSM8K | Gaokao |
|---|---|---|---|---|---|---|---|
| GPT-4o | 76.6 | 9.3 | 47.5 | 43.3 | 48.5 | 92.9 | 67.5 |
| Base 7B model | 58.8 | 0.0 | 22.5 | 21.8 | 41.6 | 91.6 | 51.7 |
| rStar-Math Round 1 | 75.2 | 10.0 | 57.5 | 35.7 | 45.4 | 90.9 | 60.3 |
| rStar-Math Round 2 | 86.6 | 43.3 | 75.0 | 59.4 | 55.6 | 94.0 | 76.4 |
| rStar-Math Round 3 | 87.0 | 46.7 | 80.0 | 61.6 | 56.5 | 94.2 | 77.1 |
| rStar-Math Round 4 | **89.4** | **50.0** | **87.5** | **65.3** | **59.0** | **95.0** | **80.5** |

**The effectiveness of self-evolution**. The impressive results in Table 3 are achieved after 4 rounds of rStar-Math self-evolved deep thinking. Table 4 details the performance across rounds, showing continuous accuracy improvements. In round 1, the main improvement comes from applying SFT to the base model. Round 2 brings a significant boost with the application of a stronger PPM in MCTS, which unlocks the full potential of System 2 deep reasoning. Notably, starting from round 2, rStar-Math outperforms GPT-4o. Rounds 3 and 4 show further improvements, driven by stronger System 2 reasoning through better policy SLMs and PPMs.

**The effectiveness of step-by-step verified reasoning trajectory**. rStar-Math generates step-by-step verified reasoning trajectories, which eliminate error intermediate steps and further expand training set with more challenging problems. To evaluate its effectiveness, we use the data generated from round 4 as SFT training data and compare it against three strong baselines: *(i)* GPT-distillation, which includes open-sourced CoT solutions synthesized using GPT-4, such as MetaMath (Yu et al., 2023b), NuminaMath-CoT (Jia LI & Polu, 2024b); *(ii)* Random sampling from self-generation, which use the same policy model (i.e., policy SLM-r3) to randomly generate trajectories; *(iii)* Rejection sampling,

where 32 trajectories are randomly sampled from the policy model, with high-quality solutions ranked by our trained ORM (appendix A.2). For fairness, we select two correct trajectories for each math problem in baseline (ii) and (iii). All SFT experiments use the same training recipe.

*Table 5.* Ablation study on the effectiveness of our step-by-step verified reasoning trajectories as the SFT dataset. We report the SFT accuracy of Qwen2.5-Math-7B fine-tuned with different datasets.

| | Dataset | MATH | AIME | Olympiad Bench | College Math |
|---|---|---|---|---|---|
| GPT-4o | - | 76.6 | 9.3 | 43.3 | 48.5 |
| GPT4-distillation (Open-sourced) | MetaMath | 55.2 | 3.33 | 19.1 | 39.2 |
| | NuminaMath-CoT | 69.6 | 10.0 | 37.2 | 43.4 |
| Self-generation by policy SLM-r3 | Random sample | 72.4 | 10.0 | 41.0 | 48.0 |
| | Rejection sampling | 73.4 | 13.3 | 44.7 | 50.8 |
| | **Step-by-step verified (ours)** | **78.4** | **26.7** | **47.1** | **52.5** |

Table 5 shows the math reasoning accuracy of Qwen2.5-Math-7B fine-tuned on different datasets. We highlight two observations: **(i)** Fine-tuning with our step-by-step verified trajectories significantly outperforms all other baselines. This is primarily due to our PPM-augmented MCTS for code-augmented CoT synthesis, which provides denser verification during math solution generation. It proves more effective than both random sampling, which lacks verification, and rejection sampling, where ORM provides only sparse verification. **(ii)** Even randomly sampled code-augmented CoT solutions from our SLM yields comparable or better performance than GPT-4 synthesized NuminaMath and MetaMath datasets. This indicates that our policy SLMs, after rounds of self-evolution, can generate high-quality math solutions. These results demonstrates the huge potential of our method to self-generate higher-quality reasoning data without relying on advanced LLM distillation.

**The effectiveness of PPM**. We train both a strong ORM and Q-value score-based PRM (PQM) for comparison. To ensure a fair evaluation, we use the highest-quality training data: the step-by-step verified trajectories generated in round 4, with selected math problems matching those used for PPM training. Similar to PPM, we use step-level Q-values as to select positive and negative trajectories for each math problem. The ORM is trained using a pairwise ranking loss (Ouyang et al., 2022), while the PQM follows (Chen et al., 2024; Zhang et al., 2024b) to use Q-values as reward labels and optimize with MSE loss. Detailed training settings are provided in Appendix A.2.

*Table 6.* Ablation study on the reward model. Process reward models (PQM and PPM) outperform ORM, with PPM pushing the frontier of math reasoning capabilities.

| RM | Inference | MATH | AIME | Olympiad Bench | College Math |
|---|---|---|---|---|---|
| o1-mini | - | **90.0** | **56.7** | **65.3** | 55.6 |
| ORM | Best-of-N | 82.6 | 26.7 | 55.1 | 55.5 |
| PQM | MCTS | 88.2 | 46.7 | 62.9 | **57.6** |
| PPM | MCTS | **89.4** | **50.0** | **65.3** | **59.0** |

Table 6 compares the performance of ORM, PQM, and PPM for System 2 reasoning using our final round policy model. ORM provides reward signals only at the end of problem solving, so we use the Best-of-N, while PQM and PPM leverage MCTS-driven search. As shown in Table 6, both PQM and PPM outperform ORM by providing denser step-level reward signals, leading to higher accuracy on complex math reasoning tasks. However, PQM struggles on more challenging benchmarks, such as MATH and Olympiad Bench, due to the inherent imprecision of Q-values. In contrast, PPM constructs step-level preference data for training, enabling our 7B policy model to achieve comparable or superior performance to o1-mini across all benchmarks.

## 5 Conclusion

In this work, we present rStar-Math, a self-evolved System 2 deep thinking approach that significantly boosts the math reasoning capabilities of small LLMs, achieving state-of-the-art OpenAI o1-level performance. Extensive experiments across four different-sized SLMs and challenging math benchmarks demonstrate the superiority of rStar-Math, with achieving leading results while outperforming existing math reasoning LLMs and Best-of-N baselines. We observe that there can be further improvements with more challenging math problems, which we leave as future work.

## Impact Statement

This paper presents work whose goal is to advance the field of deep learning. While our research focuses on enhancing the mathematical reasoning capabilities of large language models, we acknowledge potential societal implications such as applications in education and automated problem-solving. There are many broader consequences of our work, none of which we feel must be specifically highlighted here.

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

# A   Appendix

## A.1   Findings and Discussions

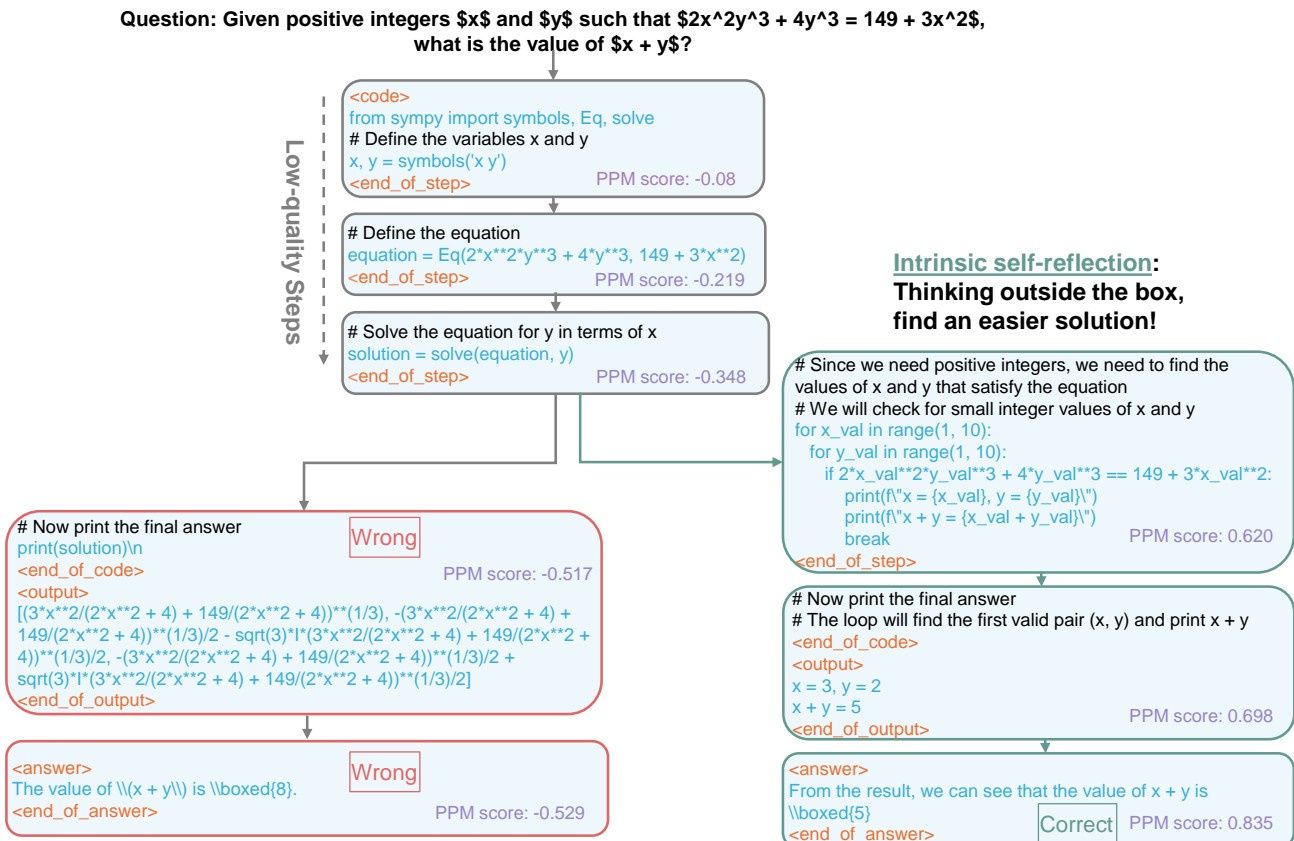

*Figure 4.* An example of intrinsic self-reflection during rStar-Math deep thinking.

**The emergence of intrinsic self-reflection capability**. A key breakthrough in OpenAI o1 is its intrinsic self-reflection capability. When the model makes an error, it recognizes the mistake and self-correct with a correct answer (Noam Brown & Lightman, 2024). Yet it has consistently been found to be largely ineffective in open-sourced LLMs. Thus, the community has actively explored various approaches, including self-correction (Huang et al., 2023; Kumar et al., 2024), self-reflection (Renze & Guven, 2024; Shinn et al., 2024), to explicitly train or prompt LLMs to develop such capability.

In our experiments, we unexpectedly observe that our MCTS-driven deep thinking exhibits self-reflection during problem-solving. As shown in Fig. 4, the model initially formalizes an equation using SymPy in the first three steps, which would lead to an incorrect answer (left branch). Interestingly, in the fourth step (right branch), the policy model recognizes the low quality of its earlier steps and refrains from continuing along the initial problem-solving path. Instead, it backtracks and resolves the problem using a new, simpler approach, ultimately arriving at the correct answer. An additional example of self-correction is provided in AppendixA.3. Notably, no self-reflection training data or prompt was included, suggesting that advanced System 2 reasoning can foster intrinsic self-reflection.

**PPM shapes the reasoning boundary in System 2 deep thinking**. Both the policy and reward models are crucial for System 2 deep reasoning. Our experiments show that different policy models achieve similar performance in System 2 reasoning (see Appendix A.2 ), with the PPM largely determines the upper performance limit. Fig. 5 summarizes the accuracy of policy models of different sizes and the accuracy achieved after applying reward models. We conclude that, despite differences in Pass@1 accuracy due to variations in training methods, data, and model size, the reward model is the key factor in System 2 reasoning. For instance, although the SFT accuracy of rStar-Math-7B is lower than Qwen2.5-Math-72B-Instruct, pairing it with our 7B PPM allows rStar-Math to outperform the 72B policy model with Qwen 72B ORM. Moreover, despite varying Pass@1 accuracy across our three policy SLM sizes, the final reasoning accuracy converges after applying the PPM.

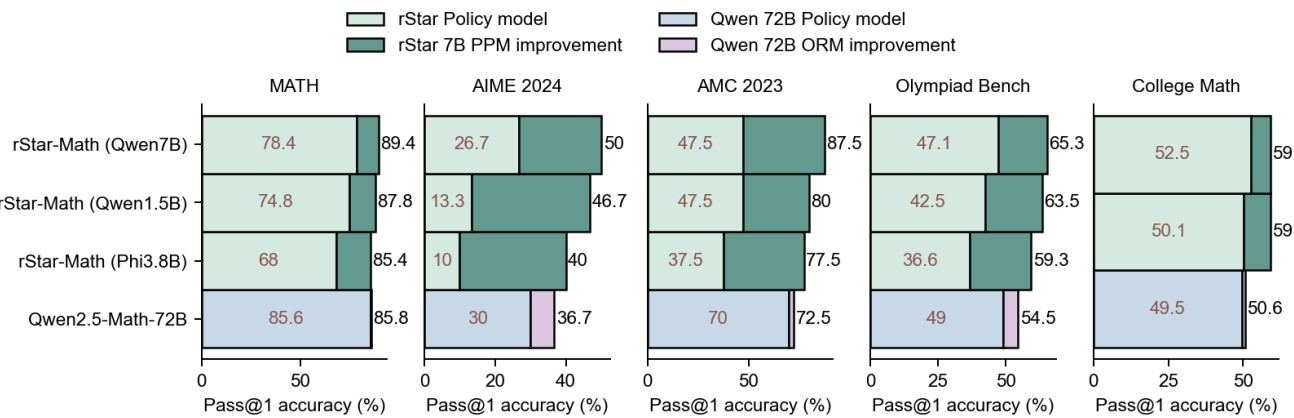

*Figure 5.* Pass@1 accuracy of policy models and their accuracy after applying System 2 reasoning with various reward models, shows that reward models primarily determine the final performance.

**PPM spots theorem-application steps**. When solving challenging math problems, identifying and applying relevant theorems or key conclusions often form the cornerstone of successful problem-solving (Xin et al., 2024). In our experiments, we find that during rStar-Math problem-solving, our PPM effectively identifies critical theorem-application intermediate steps within policy model's deep thinking process. These steps are predicted with high reward scores, guiding the policy model to the correct solution. Appendix A.3 provides examples where the PPM successfully identifies key theorems such as Fermat's little theorem (Weisstein, a), Vieta's formulas (Weisstein, b), the AM-GM inequality (amg), the Pythagorean theorem (pyt), and the Shoelace Theorem (sho), etc.

**Generalization to theorem proving**. rStar-Math is generalizable to more challenging math tasks, such as theorem proving, though its current focus is on word problems due to dataset limitations. Nonetheless, rStar-Math demonstrates the ability to prove mathematical statements. As shown in Appendix A.3, it successfully proves an Olympiad-level problem involving Fermat's Little Theorem, providing a step-by-step correct proof through its deep reasoning process. This achievement is primarily due to the policy model's strong reasoning capabilities, as the current PPM, which lacks step-level proof training, cannot yet reliably evaluate intermediate steps. Future work will focus on creating step-level proof datasets to enable the PPM to better solve theorem proving tasks.

### A.2  Additional Experiments and Details

**Data Generation Details**. As detailed in Sec. 3.3, each round starts by self-generating step-by-step verified trajectories for 747k math word problems. The maximum tree depth $d$ is set to 16, with 16 MCTS rollouts conducted per problem by default. At each step, we allow to explore 8 candidate nodes, and the constant $c$ in Eq. 1 is set to 2 to promote greater exploration. In the bootstrap round, due to the large size of the initial policy model (236B), we used smaller parameters: 8 rollouts and 5 candidate nodes per step. To improve the accuracy of solving challenging problems in round 4, we increase the number of candidate nodes to 16 and conduct 2 MCTS tree expansions per problem using different random seeds. Detailed prompts are available in Appendix A.4.

**Training Details**. In each round, we collect step-by-step verified trajectories to fine-tune the policy LLM and train the PPM. To reduce noise in synthetic math problems (e.g., incorrect ground-truth answers labeled by GPT-4), we remove synthetic problems with trajectories achieving less than 50% accuracy. Based on our extensive experiments, the policy LLM is fine-tuned from the initial base model in each round, rather than training incrementally on the model from the previous round. All policy SLMs are trained for 2 epochs with a sequence length of 4096 tokens and a batch size of 128. We use AdamW optimizer with a linear learning rate scheduler, setting the initial learning rate to 7e-6 for Qwen models, and a cosine scheduler with an initial learning rate of 5e-6 for Phi3-mini-Instruct. The PPM is trained for 1 epoch with a batch size of 512 and an initial learning rate of 7e-6.

*Table 7.* Training cost per round

|        | GPUs      | Training time |
| ------ | --------- | ------------- |
| Policy | 8×MI300X  | 20 hours      |
| PPM    | 8×MI300X  | 15 hours      |

**Training PPM**. The PPM is initialized from the fine-tuned policy model, with its next-token prediction head replaced by a scalar-value head consisting of a linear layer and a tanh function to constrain outputs to the range [-1, 1]. We filter out math problems where all solution trajectories are fully correct or incorrect. For problems with mixed outcomes, we select two positive and two negative examples for each step based on Q-values, which are used as preference pairs for training data.

**Training the ORM and PQM**. The Outcome Reward Model (ORM) and the Q-value-based Process Reward Model (PQM) share the same model architecture and training parameters with and our PPM. For ORM training, we filter trajectories from math problems containing both correct and incorrect solutions. Specifically, the two trajectories with the highest average Q-values are selected as positive examples, while the two with the lowest are chosen as negative examples. Following Qwen2.5-Math (Yang et al., 2024), we adopt the pairwise ranking loss (Ouyang et al., 2022) to optimize the ORM. To train the PQM, we follow (Chen et al., 2024) to use step-level Q-values as reward labels. Let $\mathbf{x} = x \oplus s_1 \oplus s_2 \oplus ... \oplus s_d$ be the trajectory, with annotated Q-values $\mathbf{Q} = (Q(s_1), Q(s_1), ..., Q(s_d))$ and predicted Q-values $\mathbf{Q'} = (Q'(s_1), Q'(s_1), ..., Q'(s_d))$ for each step. To stabilize PQM training, we treat each trajectory as a single training sample and predict Q-values for all steps simultaneously, rather than splitting it into individual per-step samples. Specifically, to predict the Q-value $Q'(s_i)$ for step $s_i$, PQM takes the trajectory from the question up to step $s_i$ (i.e., $x \oplus s_1 \oplus s_2 \oplus ... \oplus s_i$) as input and outputs a value between -1 and 1. We use a mean squared error (MSE) loss for PQM training:

$$\mathcal{L}_{prm}(\mathbf{x}) = \|\mathbf{Q} - \mathbf{Q'}\|^2 \tag{5}$$

*Table 8.* Inference costs of rStar-Math. We show the average number of generated tokens required to generate a trajectory for a given question.

| MATH | AIME 2024 | AMC 2023 | Olympiad Bench | College Math | GSM8K | GaokaoEn 2023 |
|---|---|---|---|---|---|---|
| 5453 | 15693 | 14544 | 7889 | 4503 | 3299 | 6375 |

*Table 9.* Pass@1 accuracy of the resulting policy SLM in each round, showing continuous improvement until surpassing the bootstrap model.

| Round# | MATH | AIME 2024 | AMC 2023 | Olympiad Bench | College Math | GSM8K | GaokaoEn 2023 |
|---|---|---|---|---|---|---|---|
| DeepSeek-Coder-V2-Instruct (bootstrap model) | 75.3 | 13.3 | 57.5 | 37.6 | 46.2 | 94.9 | 64.7 |
| Base (Qwen2.5-Math-7B) | 58.8 | 0.0 | 22.5 | 21.8 | 41.6 | 91.6 | 51.7 |
| policy SLM-r1 | 69.6 | 3.3 | 30.0 | 34.7 | 44.5 | 88.4 | 57.4 |
| policy SLM-r2 | 73.6 | 10.0 | 35.0 | 39.0 | 45.7 | 89.1 | 59.7 |
| policy SLM-r3 | 75.8 | 16.7 | 45.0 | 44.1 | 49.6 | 89.3 | 62.8 |
| policy SLM-r4 | 78.4 | 26.7 | 47.5 | 47.1 | 52.5 | 89.7 | 65.7 |

*Table 10.* The quality of PPM consistently improves across rounds. The policy model has been fixed with policy SLM-r1 for a fair comparison.

| Round# | MATH | AIME 2024 | AMC 2023 | Olympiad Bench | College Math | GSM8K | GaokaoEn 2023 |
|---|---|---|---|---|---|---|---|
| PPM-r1 | 75.2 | 10.0 | 57.5 | 35.7 | 45.4 | 90.9 | 60.3 |
| PPM-r2 | 84.1 | 26.7 | 75.0 | 52.7 | 54.2 | 93.3 | 73.0 |
| PPM-r3 | 85.2 | 33.3 | 77.5 | 59.5 | 55.6 | 93.9 | 76.6 |
| PPM-r4 | 87.0 | 43.3 | 77.5 | 61.5 | 56.8 | 94.2 | 77.8 |

**Inference Setting**. In our evaluation, we run multiple MCTS to generate candidate solution trajectories. For each problem, we generate 32 candidate nodes at each step and use the PPM to score each node. Since the PPM is effective at providing step-level quality evaluations, we did not perform multiple MCTS rollouts to update per-step Q-values. After each MCTS finsh, we select the trajectory with highest PPM score as the final answer. Table 8 presents the average number of tokens generated to produce a trajectory in MCTS.

**Self-evolution Inference Costs**. In the initial bootstrap round, we use DeepSeek-Coder-v2-Instruct (236B) as the policy model, using 10 nodes of 8×80GB H100 GPUs with 8 MCTS rollouts. This required approximately two weeks to finish the

data generation. For rounds 2–4, using our fine-tuned 7B SLM as the policy model, data generation was performed on 15 nodes of 4×40GB A100 GPUs, with each round completed in three days. In the final round, to include more challenging problems, we increased the number of MCTS rollouts to 64, extending the data generation time to one week.

*Table 11.* Pass@1 accuracy of our fine-tuned policy models for Phi3-mini, Qwen2.5-Math-1.5B, Qwen2-Math-7B and Qwen2.5-Math-7B.

| Model | MATH | AIME 2024 | AMC 2023 | Olympiad Bench | College Math | GSM8K | GaokaoEn 2023 |
|---|---|---|---|---|---|---|---|
| *General Base Model: Phi3-mini-Instruct (3.8B)* | | | | | | | |
| Phi3-mini-Instruct | 41.4 | 3.33 | 7.5 | 12.3 | 33.1 | 85.7 | 37.1 |
| **Our policy model** | **68.0** | **10.0** | **37.5** | **36.6** | **48.7** | **87.9** | **53.2** |
| *Math-Specialized Base Model: Qwen2.5-Math-1.5B* | | | | | | | |
| Qwen2.5-Math-1.5B | 51.2 | 0.0 | 22.5 | 16.7 | 38.4 | 74.6 | 46.5 |
| Qwen2.5-Math-1.5B-Instruct | 60.0 | 10.0 | **60.0** | 38.1 | 47.7 | **84.8** | 65.5 |
| **Our policy model** | **74.8** | **13.3** | 47.5 | **42.5** | **50.1** | 83.1 | **58.7** |
| *Math-Specialized Base Model: Qwen2-Math-7B* | | | | | | | |
| Qwen2-Math-7B | 53.4 | 3.3 | 25.0 | 17.3 | 39.4 | 80.4 | 47.3 |
| Qwen2-Math-7B-Instruct | 73.2 | 13.3 | **62.5** | 38.2 | 45.9 | **89.9** | 62.1 |
| **Our policy model** | **73.8** | **16.7** | 45.0 | **43.9** | **52.0** | 88.3 | **65.2** |
| *Math-Specialized Base Model: Qwen2.5-Math-7B* | | | | | | | |
| Qwen2.5-Math-7B | 58.8 | 0.0 | 22.5 | 21.8 | 41.6 | 91.6 | 51.7 |
| Qwen2.5-Math-7B-Instruct | **82.6** | 6.0 | **62.5** | 41.6 | 46.8 | **95.2** | **66.8** |
| **Our policy model** | 78.4 | **26.7** | 47.5 | **47.1** | **52.5** | 89.7 | 65.7 |

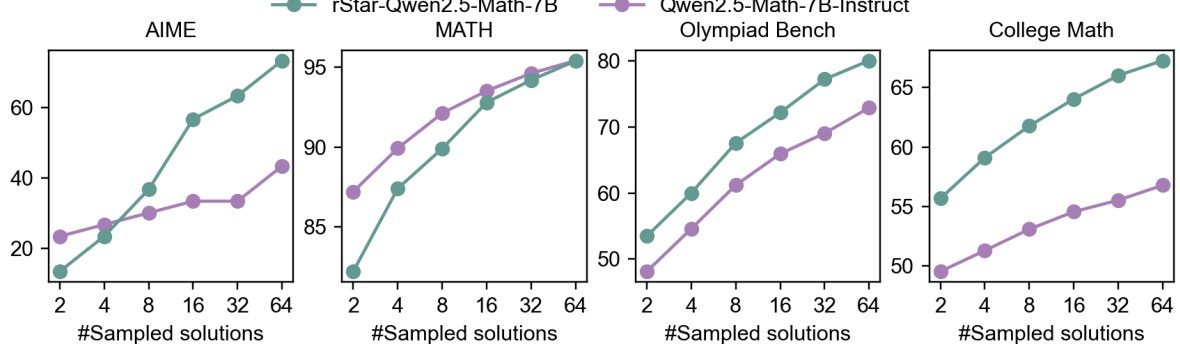

*Figure 6.* Pass@N accuracy with random sampling from different policy models. Compared to the official Qwen instruct version, our policy model exhibits a stronger ability to sample correct solutions.

**Pass@N.** Table 11 compares the math reasoning performance of our policy models with the instruct versions developed by the original model team. Our policy models do not consistently outperform the instruct versions. For example, on the Qwen2.5-Math-7B base model, Qwen2.5-Math-7B-Instruct achieves 4.2% higher accuracy on the MATH benchmark. However, the pass@1 accuracy alone does not fully reflect the reasoning capabilities for the policy model in System 2 deep thinking paradigm. To provide a more comprehensive evaluation, Fig.6 and Fig.7 present the pass@N accuracy. In this metric, the policy model generates $N$ solutions under two settings: random sampling (Fig.6) and PPM-augmented MCTS deep thinking (Fig.7). A problem is considered solved if one solution is correct. As shown in Figure 6, our policy model performs similarly to Qwen2.5-Math-7B-Instruct on the MATH benchmark's pass@64 and significantly outperforms it on others. This suggests that despite initial pass@1 accuracy differences, our policy model can generate correct solutions through multiple samples. Figure 7 further compares the pass@N accuracy of our four policy models (different sizes) after MCTS deep thinking under the same PPM guidance. We can observe that after generating 64 trajectories, the pass@N accuracy of different policy models becomes comparable.

**Ablation on the MCTS search parameters**. We perform additional analysis on the number of candidate nodes per step in MCTS. As shown in Table 12, increasing the number of candidate nodes generally improves accuracy, but beyond 32 nodes, performance saturates.

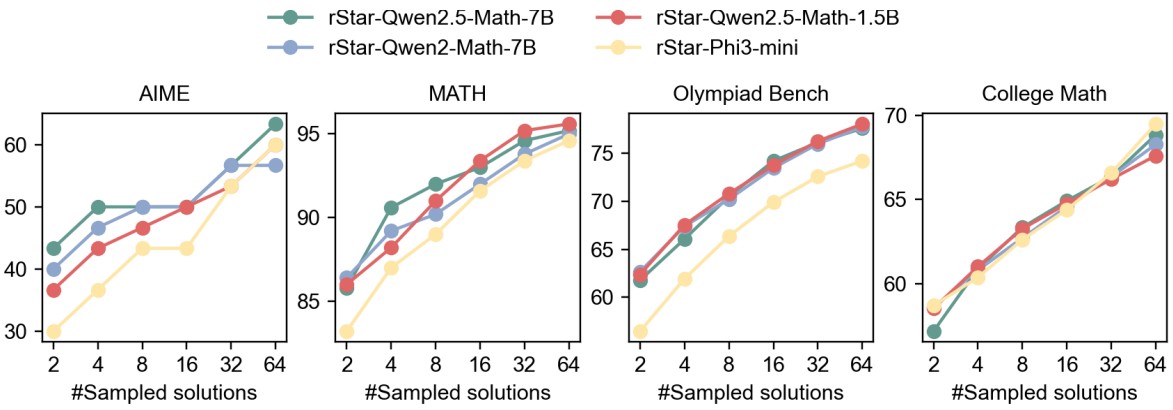

*Figure 7.* Pass@N accuracy with PPM-augmented MCTS. Under the same PPM guidance, the four policy models of varying sizes demonstrate convergent capabilities in sampling correct solutions.

*Table 12.* Ablation study on the number of candidate nodes in MCTS.

|          | MATH-500   |             | AIME 2024  |             |
|----------|------------|-------------|------------|-------------|
|          | 8 rollouts | 64 rollouts | 8 rollouts | 64 rollouts |
| node=4   | 87.2       | 88.8        | 33.3       | 36.7        |
| node=8   | 87.2       | 88.8        | 33.3       | 43.3        |
| node=16  | 88.4       | 89.0        | 36.7       | 50.0        |
| node=32  | 89.4       | 89.0        | 50.0       | 53.3        |
| node=40  | 89.4       | 90.0        | 46.7       | 53.3        |

## A.3  Examples of Reasoning Trajectories

---
**Self-correction Example**
---

Question: The coordinates of a parallelogram are $(5, 3)$, $(6, 8)$, $(7, 4)$ and $(x, y)$ and $x > 7$. What is the value of $x + y$

**Our trajectory**:
*< code >*
```
# Import necessary libraries
import numpy as np
```
*< end_of_step >*

```
# Define the coordinates of the parallelogram
A = np.array([5, 3])
B = np.array([6, 8])
C = np.array([7, 4])
```
*< end_of_step >*

```
# Calculate the vector differences to find the missing coordinate D
# In a parallelogram, the vector difference between two adjacent sides is equal to the vector difference between the other two adjacent sides
# We can use the formula: D = A + (C - B)
D = A + (C - B)
```
*< end_of_step >*

```
# Since the coordinates provided are (5, 3), (6, 8), (7, 4) and (x, y), we can infer that the vector from B to C should be the same as the vector from A to the missing point (x, y)
# This means: (x - 5, y - 3) = (1, -4) because the vector from B to C is (7 - 6, 4 - 8) = (1, -4)

# Solve for x and y
x = 5 + 1
y = 3 + (-4)
```
*< end_of_step >*

```
# Verify that the point (x, y) exists and satisfies the condition x ¿ 7
# Since x = 6 is not greater than 7, we need to use the correct formula that maintains the parallelogram property and x ¿ 7

# Correcting the formula to find the other possible D [Self-correction!]
D = C + (B - A)
```
*< end_of_step >*

```
# Solve for x and y again
x = 7 + (6 - 5)
y = 4 + (8 - 3)
```
*< end_of_step >*

```
# Verify that the point (x, y) exists and satisfies the condition x ¿ 7
# Since x = 8 is greater than 7, which satisfies the condition given, we can now find the sum of x and y
x_plus_y = x + y
```
*< end_of_step >*

```
# Now print the final answer
print(x_plus_y)
# This should print the value of x + y
```
*< end_of_code >*
*< output >*17*< end_of_output >*
*< answer >*The value of x + y is $\boxed{17}$ *< end_of_answer >*

---

**Math statement proof using Fermat's little theorem**

Question: If ( pt+a ) is a perfect square for some value of ( t ), then ( a$^{\frac{p-1}{2}}$ - 1 ) is divisible by ( p ) (where ( p ) is a prime number).

**Our trajectory for proving**:
< code >
# Step 1: Define the variables
```
from sympy import symbols, isprime, sqrt, mod_inverse
```
# Define the variables
```
p, t, a = symbols('p t a')
```
< end_of_step >

# Step 2: Assume p*t + a is a perfect square
# This means there exists an integer k such that p*t + a = k$^2$
# Rearrange to find a in terms of p, t, and k
# a = k$^2$ - p*t
< end_of_step >

# **Step 3: Use Fermat's Little Theorem**
# Fermat's Little Theorem states that if p is a prime and a is an integer not divisible by p,
then a$^{(p-1)}$ ≡ 1 (mod p)
# We need to show that a$^{((p-1)/2)}$ - 1 is divisible by p
< end_of_step >

# Step 4: Substitute a = k$^2$ - p*t into Fermat's Little Theorem
# a$^{((p-1)/2)}$ ≡ (k$^2$ - p*t)$^{((p-1)/2)}$ (mod p)
# Since p*t is divisible by p, (k$^2$ - p*t)$^{((p-1)/2)}$ ≡ k$^{(p-1)}$ (mod p)
# By Fermat's Little Theorem, k$^{(p-1)}$ ≡1 (mod p)
< end_of_step >

# Step 5: Therefore, a$^{((p-1)/2)}$ ≡ 1 (mod p)
# This implies a$^{((p-1)/2)}$ - 1 ≡ 0 (mod p)
# Hence, a$^{((p-1)/2)}$ - 1 is divisible by p
< end_of_step >

# Now print the final answer
print("The expression a$^{((p-1)/2)}$ - 1 is divisible by p.")
< end_of_code >
< output >
The expression a$^{((p-1)/2)}$ - 1 is divisible by p.
The statement is true.
< end_of_output >

**Problem-solving with application of the Vieta's formulas and the AM-GM inequality**

Question: Let $a$ be a positive real number such that all the roots of

$$x^3 + ax^2 + ax + 1 = 0$$

are real. Find the smallest possible value of $a$.

**Our trajectory for proving**:
*< code >*
# Step 1: Define the polynomial equation
from sympy import symbols, solve, Eq

# Define the variable and the parameter
x, a = symbols('x a')
*< end_of_step >[PPM Score: -0.46430]*

# Step 2: Define the polynomial equation
polynomial = x**3 + a*x**2 + a*x + 1
*< end_of_step >[PPM Score: 0.53937]*

**# Step 3: Use Vieta's formulas to relate the coefficients to the roots**
# Since the polynomial is monic (leading coefficient is 1), the sum of the roots (taken one at a time) is -a
# The sum of the products of the roots taken two at a time is a
# The product of the roots is -1
# Since all roots are real, we can denote them as r1, r2, r3
# From Vieta's formulas:
# r1 + r2 + r3 = -a
# r1*r2 + r2*r3 + r3*r1 = a
# r1*r2*r3 = -1
*< end_of_step >* [PPM Score: **0.9989**]

**# Step 4: Use the AM-GM inequality to find the minimum value of a**
# By AM-GM, (r1*r2 + r2*r3 + r3*r1) / 3 >= (r1*r2*r3)$^{(2/3)}$
# Substituting the known values:
# a / 3 >= (-1)$^{(2/3)}$
# a / 3 >= 1
# a >= 3
*< end_of_step >* [PPM Score: **0.9999**]

# Step 5: Verify if a = 3 is achievable
# If a = 3, then r1 + r2 + r3 = -3 and r1*r2 + r2*r3 + r3*r1 = 3
# We need to check if there exist real roots that satisfy these conditions
# Consider the case where r1 = r2 = r3 = -1
# Then r1 + r2 + r3 = -3 and r1*r2 + r2*r3 + r3*r1 = 3, and r1*r2*r3 = -1
# Thus, a = 3 is achievable with real roots
*< end_of_step >* [PPM Score: 1.0]

# Step 6: Print the final answer
print(3)
*< end_of_code >*
*< output >*3*< end_of_output >*
*< answer >*The smallest possible value of a is $\boxed{3}$.
*< end_of_answer >*

**Problem-solving with application the Shoelace Theorem**

Question: In the diagram, $D$ and $E$ are the midpoints of $\overline{AB}$ and $\overline{BC}$ respectively. Determine the area of quadrilateral $DBEF$. [asy]
size(180); defaultpen(linewidth(.7pt)+fontsize(10pt));
pair A, B, C, D, E, F;
A=(0,6);
B=(0,0);
C=(8,0);
D=(0,3);
E=(4,0);
F=(8/3,2);
draw(E–A–C–D);
draw((-1,0)–(10,0), EndArrow);
draw((0,-1)–(0,8), EndArrow);
label("$A(0,6)$", A, W);
label("$B(0,0)$", B, SW);
label("$C(8,0)$", C, S);
label("$D$", D, W);
label("$E$", E, S);
label("$F$", F, SW);
label("$x$", (10,0), dir(0));
label("$y$", (0,8), dir(90)); [/asy]

**Our trajectory for proving**:
*< code >*
# Step 1: Define the coordinates of points A, B, C, D, and E

```
A = (0, 6)
B = (0, 0)
C = (8, 0)
D = (0, 3) # Midpoint of AB
E = (4, 0)  # Midpoint of BC
```
*< end_of_step >[PPM Score: -0.76639]*

# Step 2: Define the coordinates of point F using the section formula (known to be (8/3, 2))
```
F = (8/3, 2)
```
*< end_of_step >[PPM Score: 0.8429]*

# **Step 3: Use the Shoelace Theorem to find the area of quadrilateral DBEF**
# Shoelace Theorem formula: Area = 0.5 * —x1y2 + x2y3 + x3y4 + x4y1 - (y1x2 + y2x3 + y3x4 + y4x1)—
# Coordinates of D, B, E, F are (0, 3), (0, 0), (4, 0), (8/3, 2) respectively
```
x1, y1 = D
x2, y2 = B
x3, y3 = E
x4, y4 = F
area = 0.5 * abs(x1*y2 + x2*y3 + x3*y4 + x4*y1
            - (y1*x2 + y2*x3 + y3*x4 + y4*x1))
```
*< end_of_step >[PPM Score: **0.9972**]*

# Step 4: Print the area of quadrilateral DBEF
```
print(area)
```
*< end_of_code >*
*< output >*8.0*< end_of_output >*
*< answer >* 8 *< end_of_answer >*

## A.4 Prompt Examples

---

**MCTS Prompt for DeepSeek-V2-Coder-Instruct (bootstrap round)**

You are a powerful agent with broad math knowledge and great python programming skills. You need to use python interpreter to do accurate calculation on math equations.

!!! Remember:
1. Use code solve the problem step by step. The solution should include three parts: $< code >$, $< output >$, and $< answer >$.
2. All calculations should be done in python code. Provide concise reasoning and thinking in the comments of the code.
3. The most related python packages include 'math', 'sympy', 'scipy', and 'numpy'.
4. Please use the following template:

Question: the input question
$< code >$Construct the code step by step. Use $< end\_of\_step >$ to indicate the end of each step. Ensure your code can execute correctly(excluding $< end\_of\_step >$) and print the answer. Avoid undefined variables (NameError), unimported packages, or formatting errors (SyntaxError, TypeError). In the last step of the code, print the final answer and add a comment: Now print the final answer.$< end\_of\_code >$
$< output >$Execute the code in using the Python interpreter and display the printed results.$< end\_of\_output >$
$< answer >$The concise answer without verbose context, put your final answer's numerical part (without unit, only focus on the numerical part if it's a choice question) in
boxed.$< end\_of\_answer >$ Now! It's your turn.
Question: **{input}**

The following are 2 demonstration examples:
Question: Terrell usually lifts two 20-pound weights 12 times. If he uses two 15-pound weights instead, how many times must Terrell lift them in order to lift the same total weight?
$< code >$
# Step 1: Calculate the total weight lifted with two 20-pound weights
total_weight_20 = 2 * 20 * 12
$< end\_of\_step >$

# Step 2: Calculate the weight lifted per repetition with two 15-pound weights
weight_per_rep_15 = 2 * 15
$< end\_of\_step >$

# Step 3: Calculate the number of repetitions needed to lift the same total weight with two 15-pound weights
reps_needed = total_weight_20 / weight_per_rep_15
$< end\_of\_step >$

# Now print the final answer
print(reps_needed)
$< end\_of\_code >$
$< output >$16.0 $< end\_of\_output > < answer >$From the result, we can see that Terrell must lift the 15-pound weights
boxed16 times to lift the same total weight.
$< end\_of\_answer >$,

Question: Find the value of $x$ that satisfies $\frac{\sqrt{3x+5}}{\sqrt{6x+5}} = \frac{\sqrt{5}}{3}$. Express your answer as a common fraction.
$< code >$
from sympy import symbols, Eq, solve, sqrt

# Define the variable x
x = symbols('x')
$< end\_of\_step >$

# Define the equation
equation = Eq(sqrt(3*x + 5) / sqrt(6*x + 5), sqrt(5) / 3)
$< end\_of\_step >$

# Solve the equation for x
solution = solve(equation, x)
$< end\_of\_step >$

# Now print the final answer
print(solution)
$< end\_of\_code >$
$< output >$[20/3]

$< end\_of\_output > < answer >$From the result, we can see that the value of x is $\boxed{\dfrac{20}{3}}$

$< end\_of\_answer >$

---

---

**MCTS Prompt for our policy SLM**

**MCTS prompt for next step generation**:
$< |user| >$:
**{input question}**
$< |assistant| >$: Let's think step by step and solve the problem with code. **{trajectory of previous steps}**

**SFT Prompt Example**:
$< |user| >$:
How many numbers can you get by multiplying two or more distinct members of the set
$1, 2, 3, 5, 11$ together?

$< |assistant| >$: Let's think step by step and solve the problem with code.
*< code >*
```
# Step 1: Create the set of numbers
numbers = {1, 2, 3, 5, 11}
```
*< end_of_step >*

```
# Step 2: Define a function to generate all possible products of two or more distinct members of the set
from itertools import combinations
def generate_products(numbers):
    products = set()
    for r in range(2, len(numbers) + 1):
        # For combinations of 2, 3, 4, and 5 elements
        for combo in combinations(numbers, r):
            product = 1
            for num in combo:
                product *= num
                products.add(product)
                return products
```
*< end_of_step >*

```
# Step 3: Calculate the products generate_products(numbers)
```
*< end_of_step >*

```
# Step 4: Calculate the number of unique products
num_unique_products = len(unique_products)
```
*< end_of_step >*

```
# Now print the final answer
print(num_unique_products)
```
*< end_of_code >*
*< output >*15*< end_of_output >*
*< answer >*From the result, we can see that there are $\boxed{15}$ unique products.
*< end_of_answer >*

---

