# OpenReview forum: "rStar-Math: Small LLMs Can Master Math Reasoning with Self-Evolved Deep Thinking"
_ICML.cc/2025/Conference — ICML 2025 oral_

### Official Review · Reviewer_xKZC · 2025-03-13

**Overall Recommendation:** 4

**Summary:**

This paper introduces rStar-Math, a novel approach demonstrating that small language models (SLMs) can achieve state-of-the-art mathematical reasoning capabilities without relying on knowledge distillation from larger models. The key innovation is a self-evolving deep thinking framework that enhances the reasoning ability of SLMs through Monte Carlo Tree Search (MCTS).

1.rStar-Math uses SLM-based policy model and reward model to iterative improves problem-solving ability through multiple rounds of self-improvement.

2.Major Innovations:
	a,Code-Augmented Chain-of-Thought Data Synthesis
	b,Process Preference Model (PPM)
	c,Four Rounds of Self-Evolution (solve increasingly difficult mathematical problems)

**Claims And Evidence:**

Questions for the claims:
rStar-Math Exhibits Intrinsic Self-Reflection Without Explicit Training:
Some examples (Figure 4) show that the model backtracks and corrects mistakes, but it is unclear if this behavior is a direct result of MCTS or an emergent property.

**Essential References Not Discussed:**

None

**Experimental Designs Or Analyses:**

Minor issues:
1.Limited theorem proving evaluation: The benchmarks focus primarily on word problems and algebraic reasoning, but lack an explicit evaluation on theorem proving, despite claims of generalization.

2.Unclear computational costs.

**Methods And Evaluation Criteria:**

The proposed methods and evaluation criteria are well-aligned with the problem of improving math reasoning in SLMs. The combination of MCTS-guided deep thinking, PPM-based process rewards, and iterative self-evolution is novel and well-validated through comprehensive benchmarking. However, high computational costs and the lack of explicit theorem-proving evaluation remain areas for improvement.

**Other Comments Or Suggestions:**

None

**Other Strengths And Weaknesses:**

Weaknesses mainly in computational efficiency and theorem proving evaluation.

**Questions For Authors:**

1.No sensitivity analysis of MCTS parameters (e.g., search depth, exploration constant)—it is unclear whether performance saturates beyond 64 rollouts.

2.Computational efficiency of self-evolution is unclear—training costs (weeks of GPU time) may be prohibitive for broader adoption.

**Relation To Broader Scientific Literature:**

1.Unlike GPT-distilled datasets, rStar-Math bootstraps its own training data via MCTS, eliminating reliance on larger models.
2.rStar-Math improves MCTS by integrating code execution validation and self-evolving data generation, leading to more reliable stepwise reasoning trajectories.
3.Unlike prior PRMs, PPM avoids noisy absolute score annotations and instead uses pairwise ranking loss, improving reward signal quality.

**Theoretical Claims:**

This is not a theory paper.

---

> ### Author Rebuttal · Authors · 2025-03-31
>
> >Q1: No sensitivity analysis of MCTS parameters—it is unclear whether performance saturates beyond 64 rollouts.
>
> **Response**: Thank you for your thoughtful review and for recognizing our contributions.
>
> We sincerely appreciate your suggestions and have conducted additional analysis on MCTS parameters, specifically focusing on the number of candidate nodes per step and the number of rollouts, which we found to be the most influential factors.
>
> 1) **number of candidate nodes per step**: To further analyze this parameter, we conducted additional experiments on Math-500 and AIME, testing different candidate node settings (4, 8, 16, 32, 40) under 8 and 64 rollouts. Notably, our paper adopts node=32. As shown in the table below, increasing the number of candidate nodes generally improves accuracy, but beyond 32 nodes, performance saturates.
>
> |MATH-500|8 rollouts| 64 rollouts|
> | :--:  | :--:   |  :--:   |
> | node=4| 87.2 | 88.8 |
> |node=8| 87.2 | 88.8 |
> |node=16| 88.4 | 89.0 |
> |node=32| 89.4 | 90.0 |
> |node=40| 89.4| 90.0|
>
> |AIME|8 rollouts| 64 rollouts|
> | :--:  | :--:   |  :--:   |
> | node=4| 33.3 | 36.7 |
> |node=8| 33.3 | 43.3 |
> |node=16| 36.7 | 50.0 |
> |node=32| 50.0 | 53.3 |
> |node=40|46.7|53.3|
>
>
> 2) **number of mcts rollouts**: As mentioned in Section 4.2 (Scaling Up Test-Time Computation), different benchmarks exhibit different trends as rollout count increases. Specifically, Math, AIME, and Olympiad benchmarks saturate at 64 rollouts. For Gaokao and College Math, where performance showed signs of further improvement beyond 64 rollouts, we conducted additional 128-rollout experiments.
>
> As shown in the following table,
> Pass@N scores consistently improve with more rollouts, but gains become marginal beyond 64 rollouts compared to the doubled search costs. On Gaokao, increasing rollouts from 64 → 128 resulted in only a slight improvement (81.3 → 81.6 for pass@1). On College Math, performance fully saturated at 128 rollouts.
>
>
> |benchmark|1| 2| 4| 8| 16| 32| 64| 128|
> | :--:  | :--:   |  :--:   | :--:   | :--:   | :--:   |  :--:   | :--:   |  :--:   |
> |Gaokao En (pass@1) | 74.5| 78.4|79.0 |80.5| 80.5|81.0 | 81.3|81.6|
> |Gaokao En (pass@n) | 74.5| 78.4| 81.8| 83.4| 85.5| 86.8| 87.0| 87.5|
> | | | | | | ||||
> |College Math (pass@1) | 55.9| 57.2| 58.0 | 59.0| 59.6| 60.1| 60.5| 60.5  |
> |College Math (pass@n)|  55.9| 57.2| 60.9| 63.5| 64.9|66.4| 67.6|68.9 |
>
>
> We appreciate the reviewer’s constructive feedback and will incorporate these analyses into our revision.
>
>
> >Q2: Computational efficiency of self-evolution is unclear—training costs (weeks of GPU time) may be prohibitive for broader adoption.
>
> **Response**: Thank you for your thoughtful feedback and for highlighting the importance of computational efficiency. We provide a detailed cost breakdown in the appendix and further clarify below.
>
> Self-evolution primarily involves two stages: (1) Policy Model & PPM Training; (2) Training Data Generation via extensive MCTS rollouts.  As shown in the following tables, training is efficient. Each round completes within a day.
>
> | |GPUs|Training time|
> | :--:  | :--:   |  :--:   |
> |Policy model|8xMI300 | 20 hours |
> |PPM | 8xMI300 | 15 hours|
>
>
> Training data generation is the main cost, but it is scalable and affordable: (1) From Round 2 onward, since our policy model and PPM are both 7B, they can be served on a single 40GB A100. (2) To process 747K math problems efficiently, we used 15 groups of 4×40GB A100s, completing data generation in ~3 days. (3) Further speedup is feasible: This process scales linearly with more GPUs (i.e., reducing the number of problems assigned per GPU). Round 1 is the only costly stage, as it requires bootstrapping with DeepSeek-Coder-V2, which was done using 8×80GB H100s.
>
>
> |Round |GPUs|Data generation time|
> | :--:  | :--:   |  :--:   |
> |Round 1|5x8x80GB H100| 2 weeks|
> |Round 2| 15x4x40GB A100| 2-3 days |
> |Round 3| 15x4x40GB A100| 2-3 days |
> |Round 4| 15x4x40GB A100| 1 week |
>
> Overall, we believe the computational cost is reasonable and manageable, and the primary bottleneck—data generation—can be further optimized with additional GPUs. We will refine our explanation in the revision to provide a clearer analysis of efficiency and scalability.

---

> > ### Comment · Reviewer_xKZC · 2025-04-04
> >
> > I really appreciate the authors' response! Overall, this is a solid paper. I hope the authors consider including their reply in the appendix—it would add valuable context and clarity.

---

> > > ### Author Response · Authors · 2025-04-07
> > >
> > > We sincerely thank you for the encouraging feedback. We will include our response in the appendix as suggested.

---

### Official Review · Reviewer_nzwK · 2025-03-13

**Overall Recommendation:** 4

**Summary:**

This paper aims to improve the mathematical reasoning capabilities of small LLMs through a self-evolved deep thinking framework, rStar-Math. The method involves three main contributions: (1) a code-augmented CoT data synthesis method; (2) a pairwise training method for the process preference model that avoids direct step-level reward annotations; and (3) a self-evolution recipe that iteratively improves the reasoning capabilities of the LLMs. Experimental results on math reasoning benchmarks show the effectiveness of the proposed method.

**Claims And Evidence:**

There is a notable inconsistency that warrants attention. The authors claim that their method achieves **self-evolved** deep thinking **without distillation** from superior models, emphasizing the independence of their approach from larger or more advanced models. This claim is central to the paper's novelty and contribution. However, in Section 3.3, Round 1, the authors state that "we run MCTS with DeepSeek-Coder-V2-Instruct (236B) to collect the SFT data". This use of a much larger model directly contradicts the claim of not relying on superior models for data synthesis. The initial bootstrap round leverages the capabilities of a 236B model, which introduces a form of distillation from a more advanced model. This inconsistency suggests an overclaim in the introduction that is not fully aligned with the methodology described later in the paper.

**Essential References Not Discussed:**

The paper has cited relevant prior work.

**Experimental Designs Or Analyses:**

The experimental designs and analyses are sound.

**Methods And Evaluation Criteria:**

- The proposed methods make sense for improving the mathematical reasoning capabilities of LLMs.
- The used math reasoning benchmarks are appropriate for assessing the performance of LLMs.

**Other Comments Or Suggestions:**

See weaknesses.

**Other Strengths And Weaknesses:**

Strengths:

- The paper is well written and easy to understand.
- The code-based cot is interesting.
- The experimental results demonstrate clear performance gains.

Weaknesses:

- The main issue with the paper is the overclaim regarding the "Self-Evolved" nature of the proposed method. The authors claim that their approach does not rely on distillation from superior models. However, the use of a 236B model (DeepSeek-Coder-V2-Instruct) in the initial round of self-evolution directly contradicts this claim. This inconsistency undermines the core novelty of the method and needs to be clarified by the authors.
- The paper proposes a code-based CoT approach to enhance math reasoning. While this method is effective for math problems, it raises concerns about domain-specificity. Specifically, the reliance on code execution and Python-based verification may limit the applicability of this approach to other domains where code-based reasoning is less relevant or feasible. The authors should address this limitation by discussing the potential generalizability of their method to other reasoning tasks beyond math.

Typo:
- In line 362, the figure number is missing.

**Questions For Authors:**

See weaknesses.

**Relation To Broader Scientific Literature:**

The key contributions of the paper are related to the broader scientific literature on improving reasoning capabilities in language models.

**Theoretical Claims:**

The paper does not introduce new theoretical contributions.

---

> ### Author Rebuttal · Authors · 2025-03-31
>
> >Q1:  The main issue with the paper is the overclaim regarding the "Self-Evolved" nature of the proposed method. The use of a 236B model (DeepSeek-Coder-V2-Instruct) in the initial round of self-evolution directly contradicts this claim.
>
> **Response**: We appreciate the reviewer' feedback regarding our claim of "self-evolved deep thinking". We clarify our approach and demonstrate that self-evolution remains the primary driver of improvement.  Notably, the effectiveness of self-evolution is acknowledged by all the other three reviewers.
>
> 1. **Clarification of self-evolution**: Our method relies on iterative improvement through MCTS-driven deep thinking. While we use DeepSeek-Coder-V2-Instruct-236B in Round 1 for bootstrapping, **this does not constitute distillation**, as the main role is to provide *an initial dataset* and plays no role in later rounds. The key novelty lies in **Rounds2-4, where we progressively train stronger 7B policy and PPM to improve independently through self-evolution**. The two models did not rely on a superior model *during the learning*, which is the key characteristics that defines model distillation. This is similar to a person who self-evolves the capability through solving problems with known answers (the initial dataset) without other people's help.
>
> 2. **Empirical evidence: performance gains from self-evolution**: Table 8 (Appendix A.2) details per-round performance.  We also provide the key results below for reference. The results show that: the policy model (round 1) did not match the performance of 236B, even with SFT data. Performance gains primarily occur in Rounds 2-4, driven by self-play and MCTS. The final 7B model outperforms the 236B model and approaches O1-mini's performance.
>
> ||MATH| AIME| Olympiad Bench| College Math|
> | :--:  | :--:   | :--:   | :--:   | :--:   |
> |DeepSeek-Coder-V2-Instruct (bootstrap model)| 75.3| 13.3|37.6|46.2|
> |our policy Round1| 69.6| 3.3|  34.7| 44.5|
> |our policy Round4| **78.4**| **26.7**| **47.1**| **52.5**|
> | | | | | |
> |o1-mini | **90.0** |**56.7** |65.3 |57.8 |
> | our policy+PPM Round4| **90.0**|53.3 |**65.6** |**60.5** |
>
> 3. **The role of round 1: code-augmented CoT format induction**: We use 236B in Round 1 primarily to ensure that our policy model can generate code-augmented CoT traces, not to transfer problem-solving ability.
>
> 4. **Distinction from Distillation** : Distillation typically involves a student model mimicking a teacher model throughout training. In contrast: (i) we use 236B only once to generate the initial dataset, (ii) self-evolution proceeds without further reliance on 236B, and (iii) later performance improvements arise purely from self-play. **Our setup is analogous to AlphaGo**, where an initial policy network is required for stability, but key performance gains stem from iterative self-evolution.
>
> To clarify, we will revise the paper to emphasize that Round 1 is for data formatting (code-augmented CoT), not capability transfer. We hope this addresses your concerns and welcome further suggestions.
>
> >Q2: The paper proposes a code-based CoT approach to enhance math reasoning. it raises concerns about domain-specificity.  The authors should address this limitation by discussing the potential generalizability of their method to other reasoning tasks beyond math.
>
> **Response**: **Clarification of Scope: focus on math reasoning**: Thank you for raising this point. We would like to emphasize that the primary focus of our paper is math reasoning, as clearly stated in the title. While generalizing to other domains is an interesting direction, exploring those is outside the scope of the current work. As you correctly point out, the potential generalizabililty of our method beyond math is definitely a worth exploring direction in our future work, as we elaborate next.
>
> 1. **Code-augmented CoT for code reasoning**: Our code-augmented CoT approach could indeed be valuable for other reasoning tasks, such as code reasoning. Code reasoning shares similarities with math reasoning in terms of structured problem-solving, making this method applicable in such domains as well.
>
> 2. **General Applicability of Self-Evolved Deep Thinking & PPM**: Beyond code-augmented CoT, our work’s core contributions — Self-Evolved Deep Thinking and the PPM — potentially offer a more general framework for various reasoning tasks. A key challenge in general reasoning lies in providing feedback to verify whether a trajectory reaches the desired outcome at the end of an MCTS rollout. In code reasoning, this could involve extensive test cases, while in other domains, feedback might come from human annotation or mutual verification with another LLM. Future work will explore these directions.
>
> Thank you again for your valuable feedback and suggestions. We hope these responses address your concerns and clarify any confusion, and we kindly ask you to consider re-evaluating our work.

---

> > ### Comment · Reviewer_nzwK · 2025-04-03
> >
> > Thanks for your clarification. I have updated my score. Good luck!

---

> > > ### Author Response · Authors · 2025-04-03
> > >
> > > Thank you for taking the time to review our responses. We truly appreciate your thoughtful feedback and your support!

---

### Official Review · Reviewer_YH6P · 2025-03-14

**Overall Recommendation:** 4

**Summary:**

This paper shows that smaller language models ($\leq7$ billion parameters) can learn to solve challenging math problems at a level comparable to much larger models (e.g., GPT-4 or o1 models). They achieve this by having these smaller models:
 - Generate and verify each step of a math solution (rather than producing it all at once).
 - Use Monte Carlo Tree Search (MCTS), guided by a specially trained “process reward model,” to explore different solution steps and pick the best paths.
 - Iteratively improve themselves (“self-evolution”) by generating new training data and refining both the small “policy” model (the one that proposes solution steps) and the reward model (the one that scores each intermediate step).

Through repeated rounds of this procedure, their 7B model eventually surpasses or matches some large commercial models (like OpenAI’s “o1-preview” or “o1-mini”) on difficult math benchmarks.

**Claims And Evidence:**

The paper’s main claims—that small models can rival larger ones in mathematical reasoning through iterative self-evolution and process-level reward models—are largely backed by systematic experiments on multiple math benchmarks, ablation studies, and comparisons to existing baselines

**Essential References Not Discussed:**

No

**Experimental Designs Or Analyses:**

The authors’ experimental design generally appears sound, particularly in their ablation studies and comparisons on multiple benchmarks, but a closer look at smaller test sets, such as the 15-problem subset of AIME, raises concerns about cherry-picking or insufficient sample size.

**Methods And Evaluation Criteria:**

Their proposed method is not as novel as stated, given that (1) the practice of augmenting solutions with comments or code already exists (e.g., program-aided language models), and (2) leveraging Monte Carlo search for data augmentation or self-evolution is a well-known technique rather than a fresh contribution. Finally, their AIME test set is very small (only 15 items), raising concerns that it could be a cherry-picked scenario.

**Other Comments Or Suggestions:**

No other comments

**Other Strengths And Weaknesses:**

A key strength is that the paper demonstrates even smaller language models can perform on par with larger ones (such as o1) when solving high school–level math problems. However, the proposed code-augmented CoT data synthesis method has two main shortcomings: while it can detect computational and symbolic deduction errors, it cannot identify logical errors (since Python typically won’t produce an exception in those cases), and its verification is limited to tasks manageable via code execution rather than general math proofs, though this may still suffice for datasets like MATH.

**Questions For Authors:**

In many RL settings, methods that rely heavily on pre-trained or trainable critic models can be easily hacked by LMs. How does your approach mitigate these risks? Could you explain any strategies you employ to ensure the policy cannot simply exploit the critic’s learned features?

**Relation To Broader Scientific Literature:**

The methods presented—such as code augmentation, step-by-step reasoning, and Monte Carlo tree search for self-evolution—are already known in the broader literature; however, this paper combines them into a cohesive pipeline. Its primary contribution lies in showing that smaller models when using these techniques, can attain performance on par with larger LLMs like o1 for certain categories of math problems.

**Theoretical Claims:**

They have no theoretical claims

---

> ### Author Rebuttal · Authors · 2025-03-31
>
> >Q1: Clarification on the evaluation benchmarks: "The authors’ experimental design generally appears sound, particularly in their ablation studies and comparisons on multiple benchmarks, but a closer look at smaller test sets, such as the 15-problem subset of AIME, raises concerns about cherry-picking or insufficient sample size."
>
> **Response**: Thank you for your thoughtful feedback! We would like to clarify that for AIME 2024, we  included both AIME I and II, resulting in 30 problems in total rather than 15. Moreover, our evaluation follows prior works [1,2,3] by selecting AIME 2024 and MATH-500 as key benchmarks, alongside six additional benchmarks, covering a total of 10,195 problems. The following table provides a detailed breakdown of the number of problems in each benchmark. Given this large-scale evaluation, we believe our results are robust and not based on cherry-picked subsets. We will revise the paper to clarify these details.
>
> |Total|MATH (500)|AIME 2024 (I&II)| AMC 2023| Olympiad Bench| College Math | GSM8K|GaokaoEn 2023| Omni-Math|
> | :--:  | :--:   |  :--:   | :--:   | :--:   | :--:   |  :--:   | :--:   | :--:   |
> |10195 |500 | 30|40|675|2818|1319|385|4428|
>
> [1] DeepSeek-R1: Incentivizing Reasoning Capability in LLMs via Reinforcement Learning
>
> [2] Kimi k1.5: Scaling Reinforcement Learning with LLMs
>
> [3] OpenAI o1, learning to reason with LLMs: https://openai.com/index/learning-to-reason-with-llms/
>
>
>
> >Q2:  the proposed code-augmented CoT data synthesis method has two main shortcomings: while it can detect computational and symbolic deduction errors, it cannot identify logical errors (since Python typically won’t produce an exception in those cases), and its verification is limited to tasks manageable via code execution rather than general math proofs, though this may still suffice for datasets like MATH.
>
> **Response**: Thank you for your insightful comment! We would like to clarify that the design of our code-augmented CoT is specifically focused on reducing computational and symbolic deduction errors, rather than logical errors. To address logical errors, we primarily rely on our proposed PPM (process preference model), which is designed to select the optimal step-level reasoning step at each stage, such that the final trajectory reaches the correct answer.
>
> Regarding math proofs,  as discussed in Appendix A.1 (Generalization to theorem proving), our method shows great potential for application to math proofs, as demonstrated by examples in Appendix A.3. The current limitation is step-level proof verification, which we believe can be addressed with a proof-capable PPM. To overcome this, our future work will focus on collecting step-level proof data to train a dedicated math proof PPM.
>
>
> >Q3: In many RL settings, methods that rely heavily on pre-trained or trainable critic models can be easily hacked by LMs. How does your approach mitigate these risks? Could you explain any strategies you employ to ensure the policy cannot simply exploit the critic’s learned features?
>
> **Response**: Thank you for your insightful question, which prompted deeper reflection and discussion on this important issue. Here are our thoughts:
>
> 1. **MCTS-driven self-evolution adopts offline updates**: Reward hacking is a known challenge in the **general RL and RLHF**, when the policy is updated online[1,2,3]. In such cases, the policy can exploit flaws in the reward model, leading to increasing reward scores without real performance gains. Our approach uses offline updates for both the policy and reward model.  In each self-evolution round, we fix the policy model and  PPM (from the previous round) and use MCTS to generate training data offline. This decoupling ensures controlled (real high-quality) data selection and significantly reduces reward hacking.
>
> 2. **Robust process-level rewards**: A key cause of reward hacking in general RL settings is the lack of reliably labeled response feedback, requiring a learned reward model for scoring. In contrast, math reasoning allows direct validation against ground truth, avoiding the need for a reward model to score final answers. To avoid potential unreliable reward label annotations during process-level reward, we incorporate: (i) MCTS Rollouts and Backpropagation: Initial step-level Q-values from the PPM are refined via MCTS rollouts, with backpropagation adjusting scores based on how often steps lead to correct answers. (ii) Process Preference Model:  Instead of using raw Q-values — often imprecise even with extensive rollouts — we construct preference pairs and train the PPM accordingly, reducing noise and improving robustness.
>
> Extensive experiments validate these strategies, showing our approach mitigates reward hacking while ensuring reliable process-level supervision. We welcome any further discussions!
>
> [1] Defining and Characterizing Reward Hacking
>
> [2] Scaling Laws for Reward Model Overoptimization
>
> [3] Reward Shaping to Mitigate Reward Hacking in RLFH

---

### Official Review · Reviewer_F2Xg · 2025-03-14

**Overall Recommendation:** 5

**Summary:**

This work presents a methodology to improve the reasoning performance for math of small language models to levels competitive with the state-of-the-art models. Specifically, the rStar-Math approach uses MCTS along with a verifier by interpreting the reasoning steps as code, in order to train a policy model and a preference model over multiple rounds. The policy model, refined by the preference model, generates reasoning trajectories, and these reasoning trajectories are used to refine the preference model, eventually leading to a policy model capable of significantly enhanced reasoning performance. Among other things, ablations show that the math data thus generated is of inherently high quality even for pure SFT.

## update after rebuttal
I maintain my strong recommendation to accept this paper to the conference and thank the authors for the interesting discussion.

**Claims And Evidence:**

I think the experiments support the claims made in this paper: the paper provides a recipe that does enable small models to be competitive with reasoning SotA of January 2025. See below for more.

**Essential References Not Discussed:**

There have been past works using code to verify steps of math reasoning before that are not mentioned/discussed in 3.1, e.g. LEVER, MathCoder (which you cite but do not discuss that aspect of it), etc.

**Experimental Designs Or Analyses:**

I think the authors set up their experiments well and ran the ablations I most immediately would have wanted them to.

**Methods And Evaluation Criteria:**

The methods make sense and the intuition is well supported by past works. The metrics used to evaluate their models are standard and appropriate. Of course, human evaluations, and analysis of efficiency vs production models would have been great to have, but can’t be expected for obvious reasons.

**Other Comments Or Suggestions:**

Typos:
1. P1: “ranking among the top 20% the brightest high school math students.” missing word
2. P3: “As a result, scaling up CoT data has diminishing returns, with gains nearing saturation.” I would slightly rewrite to make it clear that you are talking about scaling up by naively generating from frontier models, since after all your approach is also about scaling up CoT data, but in a more refined way.
3. P4: “we perform MCTS rollut”
4. P4: “we use terminal-guided annotation Formally” missing punctuation
5. Eq2: I recommend reintroducing $s_d$ (can be done by adding “$s_d$” after “Terminal nodes” in the last sentence of the paragraph following the equation)
6. P4: “PRM-augmented annotation.” should read PPM
7. Eq 4: $\sigma$ is not introduced, and if you are summing over the 4 combinations of pairwise choices of positive and negative $y_i$, it should be made clearer in the equation
8. P5: “Problems are then categories by difficulty”
9. P7: “Specifically, for AIME/AMC, we generate 16 trajectories for AIME/AMC and 8 for other benchmarks, using PPM to select the best solution.” sentencing, probably solved by removing the first “for AIME/AMC”
10. P7: “srStar-Math”
11. P7: “In Fig. ??” broken ref
12. P8: “rStar-Math can achieve further improvements by collecting more challenging math problems, we leave this as future work.” phrasing (perhaps “We observe that [there can be further improvements with more data], which we leave as future work”)
13. P13: “Training PPM” missing “the”

Suggestions:
1. P4: you evoke MCTS backpropagation. I suggest going into a bit more detail about that, perhaps in the appendix, as in my experience readers of practical LLM training methods may not be familiar with that.
2. Tab3: “Specifically, for AIME/AMC, we generate 16 trajectories for AIME/AMC and 8 for other benchmarks, using PPM to select the best solution.” I recommend adding a version of that to the table’s caption

**Other Strengths And Weaknesses:**

While this is not required in the context of the reviews, I strongly encourage the authors to share code, models and/or especially the data generated on the usual platforms the foundation model community uses if not done already.

**Questions For Authors:**

1. P3: “Candidates that execute successfully are retained as valid nodes and scored by the PPM, which assigns a Q-value” it is not clear to me why the PPM assigns the Q value as page 2 indicates that the Q value is generated separately from the number of trajectories leading to the correct answer for a given node, while the PPM is trained *from* the Q value; could you clarify that point (here and ideally either in page 2 or 3 depending on your answer)? Currently the flow of p2 and fig 1c makes it seem like the code-augmented component and Q value assignment of your pipeline happens already before (and is a prerequisite for) training the PPM.
2. Fig 2: is it intended that the python code for step 2 repeats the python code for step 1 first?
3. P5: “In each selfevolution round, we perform 16 rollouts per math problem, which leads to 16 reasoning trajectories. “ considering multiple steps and multiple future trajectories per step in rollouts, wouldn’t that lead to more than 16 reasoning trajectories? Unless you discard all but the highest scored one (which seems to be what is suggested later in the paper, in which case I’d clarify that in the paragraph).
4. Tab6: could PQM be underperforming due to PPM being more attuned to the policy model since it stems from several rounds of iterating?

**Relation To Broader Scientific Literature:**

I think this work connects well with the current interest of the community in reasoning, although see feedback below.

**Theoretical Claims:**

N/A

---

> ### Author Rebuttal · Authors · 2025-03-31
>
> Thank you for your thoughtful and positive feedback on our work. We sincerely appreciate your insights and your recognition of our contributions. Below, we address your specific comments.
>
> >Q1:  I strongly encourage the authors to share code, models and/or especially the data generated on the usual platforms the foundation model community uses if not done already.
>
> **Response**:  Thank you for your suggestion. We are committed to releasing the code and the generated data as soon as possible.
>
> >Q2: P3: “Candidates that execute successfully are retained as valid nodes and scored by the PPM, which assigns a Q-value” it is not clear to me why the PPM assigns the Q value as page 2 indicates that the Q value is generated separately from the number of trajectories leading to the correct answer for a given node, while the PPM is trained from the Q value; could you clarify that point (here and ideally either in page 2 or 3 depending on your answer)? Currently the flow of p2 and fig 1c makes it seem like the code-augmented component and Q value assignment of your pipeline happens already before (and is a prerequisite for) training the PPM.
>
> **Response**: Thank you for your question! We acknowledge that the current description may cause some ambiguity, and we will revise the paper to clarify this process.
>
> To clarify: as shown in Fig. 1(c), in the first and second rounds of data generation, we did *not* have a PPM yet. Instead, we assigned Q-values to each candidate based on terminal-guided annotation. This Q-value-labeled data was then used to train the PPM.
> Starting from the third and fourth rounds, the PPM became available, enabling PPM-augmented Q-value annotation. Specifically, during MCTS, each newly generated node is initially assigned a Q-value predicted by the PPM (whereas in the first two rounds, the initial Q-value was set to 0). This Q-value is then refined through MCTS rollouts. We will update Page 3 to make this process clearer.
>
> >Q3: Fig 2: is it intended that the python code for step 2 repeats the python code for step 1 first?
>
> **Response**: Yes, this design is to ensure that the Python code executes correctly. Specifically, Step 2's code may depend on variables or functions defined in Step 1. If a node corresponding to Step 2 were to execute only its own Python code without including the preceding steps, it could lead to syntax or execution errors due to undefined references.
>
> >Q4: P5: “In each selfevolution round, we perform 16 rollouts per math problem, which leads to 16 reasoning trajectories. “ considering multiple steps and multiple future trajectories per step in rollouts, wouldn’t that lead to more than 16 reasoning trajectories?
>
> **Response**: Thank you for your question! Our definition of rollout follows prior works [1,2,3], where a full rollout refers to a complete reasoning trajectory from the root node (question) to a terminal answer node. Thus, each rollout consists of multiple reasoning steps and explores different paths, ensuring that performing 16 rollouts per math problem yields at least 16 full reasoning trajectories. We appreciate your careful reading and will revise the paragraph to make this clearer.
>
> [1] Reasoning with language model is planning with world model, emnlp 2023
>
> [2] Mutual reasoning makes smaller LLMs stronger problem-solvers, iclr2025
>
> [3] LLaMA-Berry: Pairwise optimization for olympiad-level mathematical reasoning via o1-like monte carlo tree search
>
> >Q5: Tab6: could PQM be underperforming due to PPM being more attuned to the policy model since it stems from several rounds of iterating?
>
> **Response**: Thank you for your insightful question! The primary reason PQM underperforms compared to PPM is that using score-based Q-values as direct labels for the process reward model inherently introduces imprecision. Although extensive MCTS rollouts help improve Q-value accuracy, these Q-values struggle to differentiate fine-grained quality levels. For example, distinguishing an optimal step (score=1.0) from  a near-optimal step (score=0.9) is inherently challenging, whether through MCTS automatic annotation or human annotation. This inevitably introduces noisy scores in PQM's training data, affecting its reliability.
>
> To fully address your question, we conduct an additional experiment. We use the generated data from SLM-r1 (our first trained 7B policy SLM from round 1) to train both PQM and PPM. As shown in the table, PPM consistently outperforms PQM from the early stages of self-evolution. This empirically demonstrates that PPM's advantage stems from its superior design, which effectively mitigates the impact of noisy Q-value scores.
>
> ||MATH|AIME 2024| AMC 2023| Olympiad Bench| College Math | GaokaoEn 2023|
> | :--:  | :--:   |  :--:   | :--:   | :--:   | :--:   |  :--:   |
> |SLM-r1+PQM (trained by SLM-r1 generated data)| 82.4 | 23.3|**75.0** |49.6|52.9|70.3|
> |SLM-r1+PPM (trained by SLM-r1 generated data)| **84.0**| **26.7**| **75.0**| **52.7**| **54.2**| **73.0**|

---

> > ### Comment · Reviewer_F2Xg · 2025-04-06
> >
> > I thank the authors for their detailed answer and updating paper with these clarifications / experiments / corrections of typos.

---

> > > ### Author Response · Authors · 2025-04-07
> > >
> > > We greatly appreciate your kind acknowledgment. We will update the paper accordingly. Thank you for your valuable feedback!

---

### Decision · Program_Chairs · 2025-05-01

**Decision:**

Accept (oral)

**Comment:**

This paper introduces rStar-Math, a novel training pipeline that enables small language models (SLMs) to achieve math reasoning capabilities comparable to OpenAI’s o1. The key contributions of the proposed framework include:

1. A novel code-augmented CoT data synthesis method,
2. An innovative process reward model training method, and
3. A self-evolution recipe for iterative model improvement.

The experimental results are highly impressive. All reviewers acknowledged the significance of the paper’s contributions. While one reviewer initially raised concerns about potential overclaims, these were thoroughly addressed in subsequent discussions. The authors engaged constructively with the reviewers and resolving all questions.

Given the paper’s substantial contributions and the strong consensus among reviewers, I believe this work is highly relevant to current research interests in the community and will stimulate further discussion. Therefore, I recommend accepting this paper.